# GABA neurons in the ventral tegmental area regulate non-rapid eye movement sleep in mice

Srikanta Chowdhury[1,2,3†], Takanori Matsubara[1,2,3], Toh Miyazaki[1,2,3,4], Daisuke Ono[1,2,3], Noriaki Fukatsu[1,2], Manabu Abe[5], Kenji Sakimura[5], Yuki Sudo[6], Akihiro Yamanaka[1,2,3]*

[1]Department of Neuroscience II, Research Institute of Environmental Medicine, Nagoya University, Nagoya, Japan; [2]Department of Neural Regulation,Graduate School of Medicine, Nagoya University, Nagoya, Japan; [3]CREST, JST, Honcho Kawaguchi, Saitama, Japan; [4]Research Fellowship for Young Scientist, Japan Society for the Promotion of Science, Tokyo, Japan; [5]Department of Animal Model development, Brain Research Institute, Niigata University, Niigata, Japan; [6]Division of Pharmaceutical Sciences, Graduate School of Medicine, Dentistry, and Pharmaceutical Sciences, Okayama University, Okayama, Japan

*For correspondence:
yamank@riem.nagoya-u.ac.jp

Present address: †Department of Biochemistry and Molecular Biology, University of Chittagong, Chittagong, Bangladesh

**Abstract** Sleep/wakefulness cycle is regulated by coordinated interactions between sleep- and wakefulness-regulating neural circuitry. However, the detailed mechanism is far from understood. Here, we found that glutamic acid decarboxylase 67-positive GABAergic neurons in the ventral tegmental area ($VTA_{Gad67+}$) are a key regulator of non-rapid eye movement (NREM) sleep in mice. $VTA_{Gad67+}$ project to multiple brain areas implicated in sleep/wakefulness regulation such as the lateral hypothalamus (LH). Chemogenetic activation of $VTA_{Gad67+}$ promoted NREM sleep with higher delta power whereas optogenetic inhibition of these induced prompt arousal from NREM sleep, even under highly somnolent conditions, but not from REM sleep. $VTA_{Gad67+}$ showed the highest activity in NREM sleep and the lowest activity in REM sleep. Moreover, $VTA_{Gad67+}$ directly innervated and inhibited wake-promoting orexin/hypocretin neurons by releasing GABA. As such, optogenetic activation of $VTA_{Gad67+}$ terminals in the LH promoted NREM sleep. Taken together, we revealed that $VTA_{Gad67+}$ play an important role in the regulation of NREM sleep.

## Introduction

Sleep or sleep-like behavioral quiescence is known to be one of the most ubiquitously observed phenomena across the animal kingdom, from nematodes to primates (*Joiner, 2016*; *Siegel, 2008*). Broadly, sleep consists of non-rapid eye movement (NREM) sleep and REM sleep in mammals (*Siegel, 2008*). While the physiological functions of either NREM sleep or REM sleep, or sleep as a whole, are intriguing and shrouded in mystery, sleep deprivation in humans and experimental animals causes severe cognitive impairment (*Siegel, 2008*). Pioneering studies discovered certain physiological functions of sleep that include clearing metabolic waste products and toxins from the brain (*Xie et al., 2013*), memory encoding, consolidation and erasure (*Rasch and Born, 2013*), synaptic homeostasis (*Bushey et al., 2011*), and energy conservation (*Schmidt, 2014*). However, a universal function of sleep that is relevant to all animals is yet to be revealed (*Joiner, 2016*). As animals remain largely isolated from sensory processing and goal-oriented activity during sleep, it is expected that the regulation of sleep, both NREM and REM, as well as arousal should be controlled by the central nervous system. Many brain areas and residing cellular subtypes have been shown to be critical in regulating sleep-wakefulness. For instance, orexin/hypocretin-producing neurons

(orexin neurons) in the lateral hypothalamus (LH) project to and activate monoaminergic, cholinergic, and other peptidergic neurons as well as other orexin neurons to induce and maintain wakefulness (*Brown et al., 2012*; *Inutsuka and Yamanaka, 2013*; *Sakurai, 2007*; *Scammell et al., 2017*). Subsequently, these monoamine neurons inhibit sleep-active γ-aminobutyric acid (GABA)-ergic neurons in the ventrolateral preoptic area (VLPO) in the hypothalamus to induce wakefulness (*Saito et al., 2018*; *Saper et al., 2010*). It is reported that some wake-active neurons also display activity during REM sleep (*Brown et al., 2012*; *Scammell et al., 2017*). Comparatively, NREM sleep is regulated by neurons that release classical fast neurotransmitters, including GABA. For example, circadian rhythms and/or homeostatic sleep pressures activate GABAergic neurons in the VLPO and median preoptic nucleus (MnPO), which in turn inhibit wake-promoting orexin/hypocretin, monoaminergic, and cholinergic systems (*Scammell et al., 2017*). While this flip-flop switch model of sleep-wake regulation is well established, recent studies have demonstrated a critical involvement of other brain areas and neuronal subtypes in regulating the transformations and subsequent maintenance of specific vigilances states (*Liu et al., 2017*; *Oishi et al., 2017b*).

Reinforcement learning, motivation, and locomotion, as well as the adaptation of responses to salient stimuli, all of which demand behavioral arousal, are critically regulated by a midbrain structure called the ventral tegmental area (VTA) in both rodents and primates (*Arsenault et al., 2014*; *Fields et al., 2007*). While one could also expect a critical role for VTA in the regulation of sleep/wakefulness, it is only recently that VTA dopamine (DA) neurons have been reported to have a fundamental role in the maintenance of the awake state as well as in the consolidation of arousal in mice (*Eban-Rothschild et al., 2016*; *Oishi et al., 2017a*). However, the VTA contains considerable heterogeneity among the neuronal subtypes, which include GABAergic and glutamatergic neurons alongside DA neurons. Studies have reported that about 60–65% of VTA neurons are dopaminergic, whereas 30–35% are GABAergic, and 2–3% are glutamatergic neurons (*Nair-Roberts et al., 2008*; *Pignatelli and Bonci, 2015*).

As GABAergic neurons provide strong inhibition to the wake- and REM-active DA neurons in the VTA (*Eban-Rothschild et al., 2016*; *Tan et al., 2012*; *van Zessen et al., 2012*), it is probable that these GABAergic neurons in the VTA may also participate in sleep/wakefulness regulation. Moreover, GABA-mediated responses have been implicated in the modulation of the sleep/wakefulness cycle (*Brown et al., 2012*; *Scammell et al., 2017*). Here, we examined the role of glutamic acid decarboxylase 67 (Gad67)-positive neurons in the VTA on sleep/wakefulness by using AAV-aided whole-brain anterograde tracing, neural manipulations by chemo- and optogenetics, fiber photometry, as well as slice electrophysiology. Gad67 (encoded by *Gad1* gene) is an isomer of an enzyme that synthesizes GABA from glutamic acid, suggesting that Gad67+ neurons are GABAergic neurons (*Erlander et al., 1991*). We revealed that Gad67+ neurons in the VTA (VTA$_{Gad67+}$ neurons) are highly active during NREM sleep and send their axons to multiple brain areas that were previously reported to regulate sleep/wakefulness. Bidirectional manipulations of neuronal activity and fiber photometry recordings revealed that VTA$_{Gad67+}$ neurons are active in and promote NREM sleep. Part of the NREM sleep-promoting effect of VTA$_{Gad67+}$ is mediated through inhibition of wake-promoting orexin/hypocretin neurons in the LH.

## Results

### GABAergic neurons in the VTA project to brain areas involved in the regulation of sleep/wakefulness

Glutamatergic, GABAergic, and dopaminergic (DA) neurons are intermingled in the VTA (*Nair-Roberts et al., 2008*; *Pignatelli and Bonci, 2015*). Here, we focused on the GABAergic neurons and tried to identify relevant projection areas. To specifically target GABAergic neurons in the VTA (VTA$_{GABA}$ neurons), Gad67-Cre mice (*Higo et al., 2009*) were unilaterally injected with a Cre-inducible AAV virus carrying humanized renilla green fluorescent protein (hrGFP) (*Figure 1a*). Many hrGFP-positive neurons were observed in the VTA (*Figure 1b–d*). These hrGFP-positive neurons were Gad67-positive but tyrosine hydroxylase-negative (TH, an enzyme and marker of DA neurons in the VTA) (*Figure 1b*, n = 4 mice) confirming that these hrGFP-positive neurons were GABAergic. At least 3 weeks after unilateral injection of AAV aimed at the VTA to express hrGFP, we prepared coronal brain sections at 40 μm thickness and counted all labeled cells on every fourth section. Within

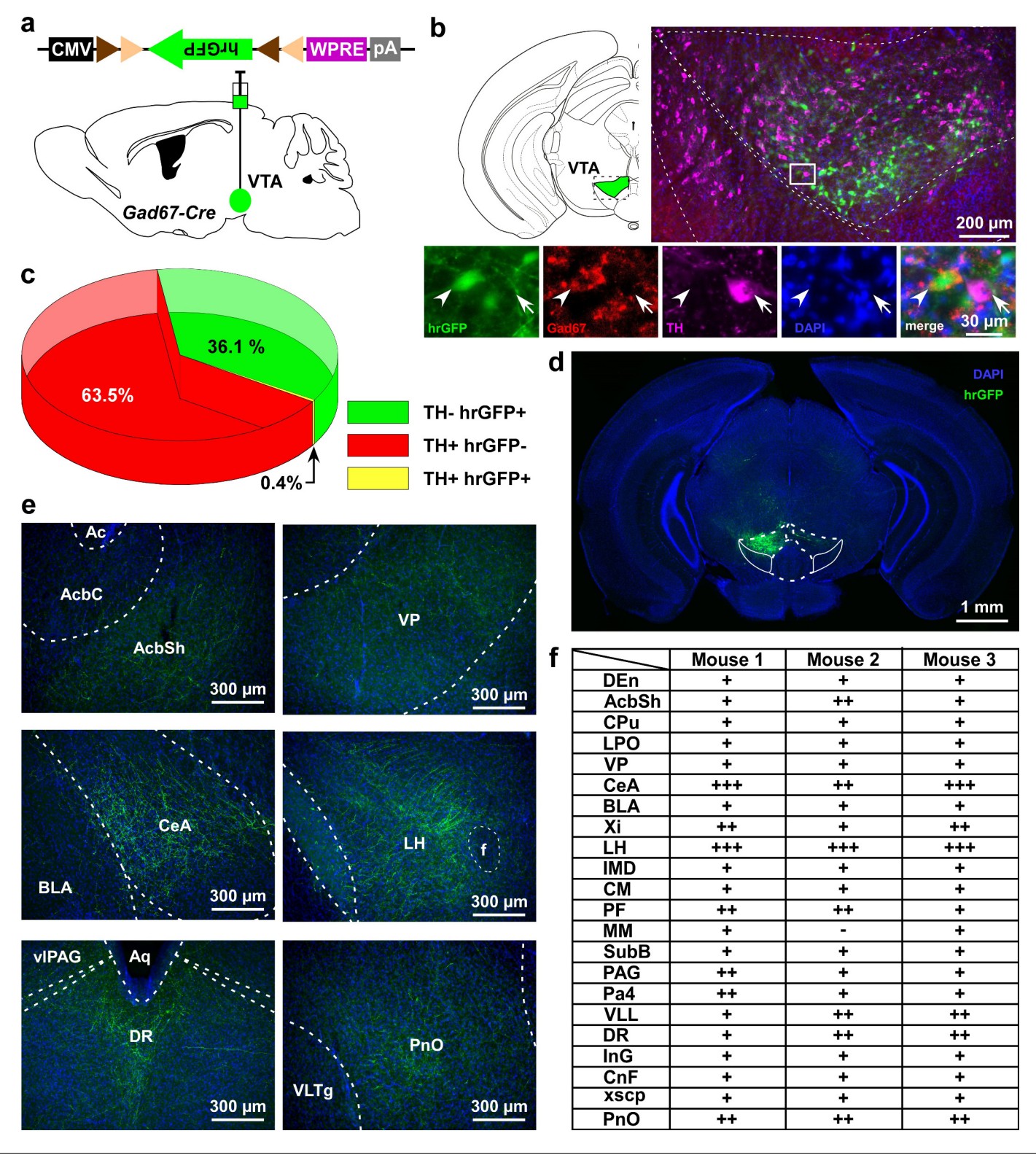

**Figure 1.** VTA_Gad67+ neurons project to multiple areas in the brain. (**a**) Schematic of AAV injection to express Cre-inducible hrGFP in Gad67-Cre mice. The dotted brain map area is shown to the right. White rectangular area is shown below. (**b**) Immunohistochemical studies showing expression of hrGFP in Gad67+ neurons (arrowhead), but not in the nearby DA (arrow) neurons. (**c**) Pie chart showing the percent of hrGFP expression in DA and non-DA neurons in the VTA (n = 4 mice). (**d and e**) Expression of hrGFP in VTA_Gad67+ neurons and some of their projected brain areas are shown. (**f**) Table showing the comparative scoring of hrGFP signals across different brain areas. Abbreviations- AcbC: Nucleus accumbens core, AcbSh: Nucleus

*Figure 1 continued on next page*

accumbens shell, BLA: Basolateral amygdala, CeA: Central nucleus of the amygdala, CM: Central medial thalamic nucleus, CnF: Cuneiform nucleus, CPu: Caudate putamen (striatum), DEn: Dorsal endopiriform nucleus, DR: Dorsal raphe nucleus, IMD: Intermediodorsal thalamic nucleus, InG: Intermediate gray layer of the superior colliculus, LH: Lateral hypothalamus, LPO: Lateral preoptic area, MM: Medial mammillary nucleus, Pa4: Paratrochlear nucleus, PAG: Periaqueductal gray, PF: Parafascicular thalamic nucleus, PnO: Pontine reticular nucleus, SubB: Subbrachial nucleus, VLL: Ventral nucleus of the lateral lemniscus, VP: Ventral pallidum, Xi: Xiphoid thalamic nucleus and xscp: Decussation of the superior cerebellar peduncles. DAPI (4',6-diamidino-2-phenylindole) staining was used to label nuclear DNA and also to assist understanding of the anatomical position of VTA$_{Gad67+}$ neuronal projections.

The online version of this article includes the following source data and figure supplement(s) for figure 1:

**Source data 1.** Source data for *Figure 1c*.
**Figure supplement 1.** Gad67+ neurons represent a sub-population of GABAergic neurons in the VTA.
**Figure supplement 1—source data 1.** Source data for *Figure 1—figure supplement 1c and d*.

the VTA, we counted a total of 636 ± 122 neurons per animal (n = 4 mice). Among them, 63.5 ± 1.8% were TH-positive neurons (DA neurons) and 36.1 ± 1.8% were hrGFP-positive neurons. Only 0.4 ± 0.1% were co-labeled with hrGFP and TH (*Figure 1c*). hrGFP was distributed not only in the soma, but also in the axons. We could even anterogradely trace axons to reveal projection sites since hrGFP emits a strong fluorescence (*Figure 1d–f*). Along with local innervations, we found long-range projections of Gad67+ neurons in the VTA (VTA$_{Gad67+}$ neurons) throughout the brain. Among these sites, the lateral hypothalamus (LH) and the central nucleus of the amygdala (CeA) were densely innervated (*Figure 1e–f*). Moderate projections were found in the nucleus accumbens (NAc), ventral pallidum (VP), parafascicular thalamic nucleus (PF), periaqueductal gray (PAG), ventral nucleus of the lateral lemniscus (VLL), dorsal raphe nucleus (DR), and pontine reticular nucleus (PnO). These brain areas are also reported to be involved in the modulation of sleep/wakefulness, suggesting that VTA$_{Gad67+}$ neurons might play a role in this regulation (*Brown et al., 2012*).

## VTA$_{Gad67+}$ neurons represent a small subset of Vgat-expressing neurons in the VTA

To reveal what percent of total GABAergic neurons in the VTA are Gad67-positive, we performed in situ hybridization using RNAscope technology. We injected Cre-inducible AAV (AAV(9)-CAG-FLEX-mCherry) into the VTA of Gad67-Cre mice (n = 4 mice) to label Gad67-expressing neurons (*Figure 1—figure supplement 1a*). We selected probes to visualize Gad67, Vgat (*SLC32A1*), and mCherry (*Table 1*). Gad67, Vgat (vesicular GABA transporter), and mCherry mRNA were visualized by multicolor in situ hybridization (*Figure 1—figure supplement 1b*). Regarding Gad67 and Vgat expression, we found three different types of neurons: Vgat-only (66.8 ± 2.0%), Gad67-only (1.1 ± 0.1%), and Vgat and Gad67 double-positive neurons (32.1 ± 2.1%) (*Figure 1—figure supplement 1c*). Whereas 96.5 ± 0.5% of Gad67-positive neurons co-expressed Vgat mRNA, only 32.4 ± 2.1% of Vgat-positive neurons co-expressed Gad67 mRNA in the VTA (*Figure 1—figure supplement 1c*). Consistent with these findings, 94.8 ± 1.5% of mCherry-positive neurons in the VTA of Gad67-Cre mice co-expressed Gad67 and Vgat mRNA (*Figure 1—figure supplement 1d*). These results indicate that Gad67-positive neurons represent a small subset of Vgat-positive neurons in the VTA.

## Chemogenetic activation of VTA$_{Gad67+}$ neurons induced NREM sleep with high delta power

Next, to reveal whether VTA$_{Gad67+}$ neurons contribute to the regulation of sleep/wakefulness, we activated these neurons by means of pharmacogenetics (chemicogenetics), using designer receptors

**Table 1.** Details of probes designed for in situ hybridization.

| Gene | Channel | Color | Position | Accession | Catalog number |
|------|---------|-------|----------|-----------|----------------|
| *Gad67* | 1 | Opal 520 | 62-3113 | NM_008077.4 | 400951 |
| *Vgat* (SLC32A1) | 4 | Opal 690 | 894-2037 | NM_009508.2 | 319191-C4 |
| *mCherry* | 3 | Opal 620 | 23-681 | n/a | 431201-C3 |

exclusively activated by designer drugs (DREADD). We bilaterally injected a Cre-inducible AAV virus to express either hM3Dq–mCherry or mCherry into the VTA of Gad67-Cre mice (*Figure 2a–b*, *Figure 2—figure supplement 1a–b*). We then confirmed the function of hM3Dq by applying its ligand clozapine-N-oxide (CNO) to acute brain slices while recording neuronal activity (*Figure 2—figure supplement 1c*). As expected, CNO application significantly increased the firing frequency of hM3Dq-expressing, but not mCherry-expressing, VTA$_{Gad67+}$ neurons (*Figure 2—figure supplement 1d–e*, hM3Dq: 286 ± 61%, n = 8 cells; mCherry: 110 ± 8%, n = 6 cells, p=0.02, unpaired *t*-test). Next, to analyze the effect of CNO-induced activation of VTA$_{Gad67+}$ neurons in sleep/wakefulness states, electroencephalogram (EEG) and electromyogram (EMG) electrodes were implanted in Gad67-Cre mice (*Figure 2a*). After recovery from the surgery and behavioral habituation (see Materials and methods), either saline or CNO (1 mg/kg) were administered intraperitoneally (i.p.) just before the onset of the dark period (at 8:00 pm). CNO administration resulted in a significantly reduced time spent in wakefulness and increased time spent in NREM sleep (also known as slow-wave sleep) in the hM3Dq-mCherry expressing mice, but not in mCherry-expressing mice (*Figure 2c–d*, hM3Dq: n = 6 mice, mCherry: n = 4 mice). The CNO-induced increase in NREM sleep lasted for at least 4 hr after CNO administration (*Figure 2d*, % change from saline in hM3Dq-mCherry-expressing mice: wakefulness 22 ± 2, NREM 259 ± 18, REM 30 ± 8, vs saline NREM, p=3.0e-4, paired *t*-test). Interestingly, the delta power (1–5 Hz) during NREM sleep was significantly increased in the CNO-injected group compared to the NREM sleep in the saline-injected control group (mean relative delta power for 4 hr post-injection: hM3Dq-saine: 83 ± 3%, hM3Dq-saine: 138 ± 8%, p=0.001, paired *t*-test), suggesting that VTA$_{Gad67+}$ neurons might be a critical regulator of slow-wave in NREM sleep (*Figure 2e–g*). However, time spent in REM sleep remained unaffected during activation of VTA$_{Gad67+}$ neurons, suggesting that VTA$_{Gad67+}$ neurons are involved in the regulation of NREM sleep, but not REM sleep (*Figure 2c–d*).

## Optogenetic inhibition of VTA$_{Gad67+}$ neurons induced wakefulness

Since activation of VTA$_{Gad67+}$ neurons resulted in increases in NREM sleep with increases in delta wave power, we next examined the selective inhibition of VTA$_{Gad67+}$ neurons, which might be expected to increase wakefulness. To test this, we used an acute inhibition strategy with optogenetics. An inhibitory anion channel, anion channelrhodopsin-2 (ACR2, Genbank accession no. KP171709) (*Mohammad et al., 2017*), was expressed in VTA$_{Gad67+}$ neurons (*Figure 3a* and *Figure 3—figure supplement 1a–b*). We first confirmed the function of ACR2 employing in vitro electrophysiology. Three weeks after injection of AAV (expressing either ACR2-2A-mCherry or mCherry) into the VTA of Gad67-Cre mice, we prepared acute brain slices including the VTA and performed cell-attached recordings from mCherry-expressing neurons. Blue light (6.8 mW/mm$^2$) was able to completely silence the spontaneous activity of ACR2-2A-mCherry-expressing VTA$_{Gad67+}$ neurons (n = 10 cells), whereas light irradiation on mCherry alone-expressing neurons had no such effect (n = 7 cells) (*Figure 3—figure supplement 1d–f*). Next, using these two groups of mice, we implanted fiber optics at a diameter of 400 µm into the VTA along with EEG and EMG electrodes (*Figure 3a and b*). After recovery and habituation, continuous blue light for 5 s was illuminated every 15 min for 24 hr (*Figure 3c*). Interestingly, blue light illumination immediately induced wakefulness from NREM sleep, but not from REM sleep, in mice expressing ACR2-2A-mCherry (n = 6 mice, *Figure 3d–f*, *Video 1*). No such effect was observed in mice expressing mCherry alone (n = 5 mice, *Video 2*). However, as the light-induced influences on NREM sleep and wakefulness showed an extended effect after the cessation of light (*Figure 3d–e*), with behaviors taking around 60 s to return to the basal state, we sought to identify whether optogenetic inhibition of VTA$_{Gad67+}$ neurons also causes prolonged wakefulness. We, therefore, isolated the trials depending on sleep-wakefulness states just before light illumination in the cases of wakefulness, NREM, or REM sleep (with the same state lasting ≥30 s before light illumination). Surprisingly, we found that optogenetic inhibition of VTA$_{Gad67+}$ neurons in the state of wakefulness prolonged the time spent in wakefulness in all sorted trials compared to behavior of the control group (*Figure 3f*, ACR2: 35 ± 5 s, mCherry: 8 ± 8 s; p=0.02, unpaired *t*-test). Again, REM sleep was not affected. Therefore, these data showed that in vivo optogenetic inhibition of VTA$_{Gad67+}$ neurons promoted and sustained wakefulness in mice. This result clearly suggested that VTA$_{Gad67+}$ neurons have a role in the regulation of not only NREM sleep but also wakefulness.

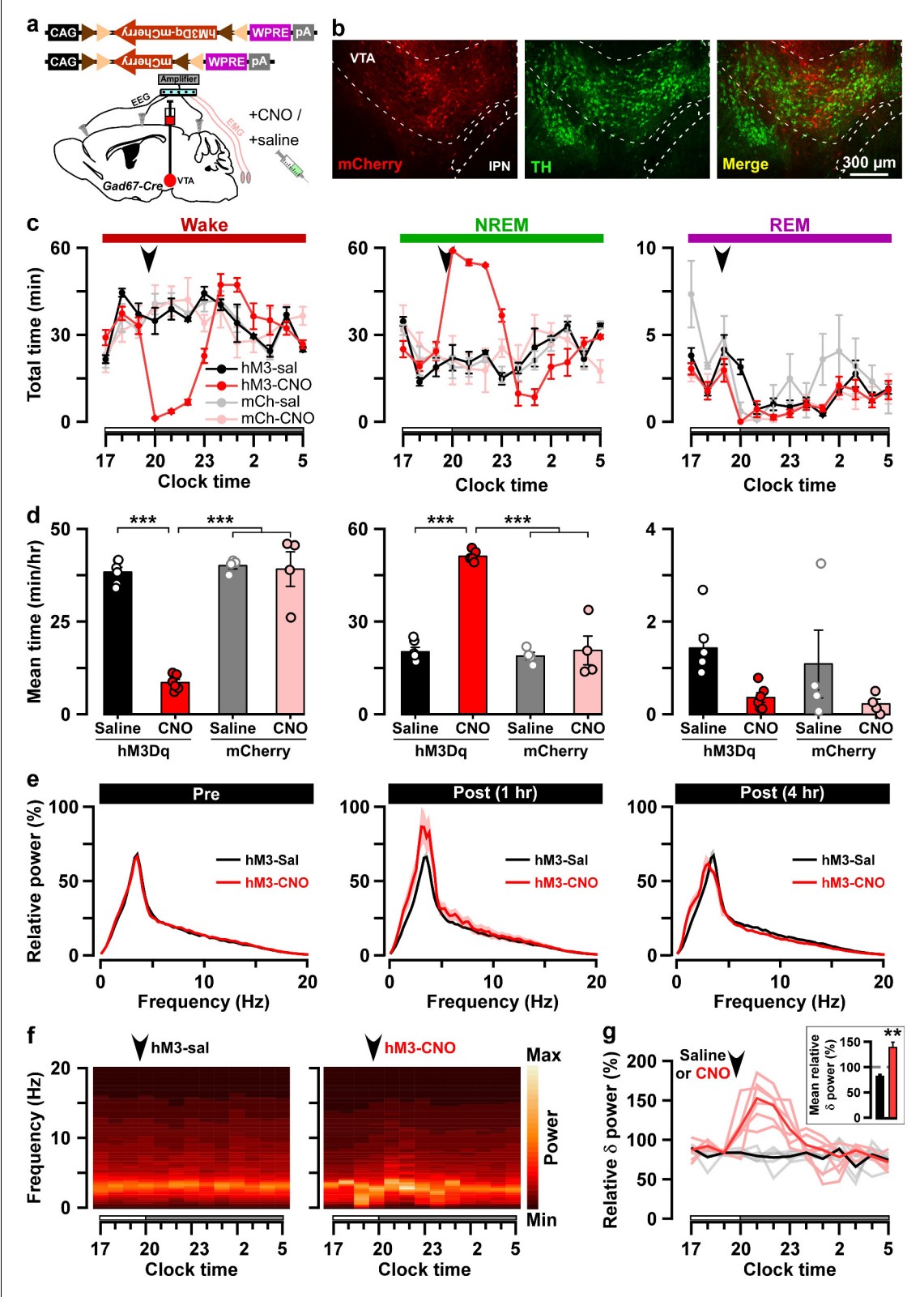

**Figure 2.** Chemogenetic activation of VTA$_{Gad67+}$ neurons induced long-lasting NREM sleep. (a) Schematic of Cre-inducible expression of either hM3Dq-mCherry or mCherry in VTA$_{Gad67+}$ neurons. (b) Immunohistochemical confirmation of hM3Dq-mCherry expression in VTA non-DA neurons. (c) Time spent in each vigilance state before and after i.p. administration of either saline or CNO. Arrowhead indicates timing of injection (just before the dark period; hM3Dq: n = 6 mice; mCherry: n = 4 mice). White and gray bars above the x-axis indicate light and dark periods, respectively. (d) 4 hr

*Figure 2 continued on next page*

Figure 2 continued

average time spent in each vigilance state after i.p. administration. (e) Relative power of fast Fourier transformation (FFT) analysis of NREM sleep for hM3Dq-expressing saline and CNO groups before (pre: left) and after (1 hr post: middle, 4 hr post: right) i.p. administration. (f) Heatmap showing a trace indicating that delta wave power activity increases after CNO administration compared to the saline control. (g) Summary of the delta wave power change during NREM sleep after saline or CNO injection. Traces in dark color indicate mean value, while lighter color indicates EEG spectrum of each mice injected with saline (black) and CNO (red). Inset shows the mean relative delta power for 4 hr after either saline or CNO administration. Data are shown as the mean ± SEM (hM3Dq: n = 6 mice; mCherry: n = 4 mice). *p<0.05, ***p<0.001, (d) two-way ANOVA followed by Tukey post hoc, (e) two-tailed paired Student's *t*-test (n = 8 mice).

The online version of this article includes the following source data and figure supplement(s) for figure 2:

**Source data 1.** Source data for *Figure 2c-e and 2g*.
**Figure supplement 1.** Histological and electrophysiological confirmation of the expression region and function of hM3Dq in the Gad67-Cre mice used in *Figure 2*.
**Figure supplement 1—source data 1.** Source data for *Figure 2—figure supplement 1e*.
**Figure supplement 2.** Relative power of FFT analysis during Wake and REM sleep for hM3Dq-expressing saline and CNO groups.
**Figure supplement 2—source data 1.** Source data for *Figure 2—figure supplement 2a and b*.

Next, we tested whether brief (5 s) optogenetic inhibition of VTA$_{Gad67+}$ neurons can induce arousal even under conditions of high homeostatic sleep pressure. To test this, mice were sleep-deprived for 4 hr, starting at light onset, and were then allowed to experience recovery sleep for 30 min (*Figure 4a*). Sleep-deprived animals usually display extended NREM sleep because of high homeostatic sleep pressure. Moreover, the slow-wave activity in NREM sleep increases during recovery sleep (*Lancel et al., 1992*). However, to our surprise, even under such a higher sleep pressure condition, optogenetic inhibition of VTA$_{Gad67+}$ neurons could successfully and immediately induce wakefulness in all trials (*Figure 4b and c*). Once again, induced-wakefulness displayed an extended effect after cessation of light, whereby it took 54 ± 14 s to return to NREM sleep. Taken together, these results suggest that VTA$_{Gad67+}$ neurons might be involved in the initiation and maintenance of physiological NREM sleep.

## VTA$_{Gad67+}$ neurons showed the highest population activity during NREM sleep

Our chemogenetic activation and optogenetic inhibition studies suggest that the in vivo activity of VTA$_{Gad67+}$ neurons might change across brain states with putatively higher activity during NREM sleep. To test this hypothesis, we recorded the population activity of VTA$_{Gad67+}$ neurons using fiber photometry (*Inutsuka et al., 2016*). A Cre-inducible AAV expressing the fluorescent calcium indicator GCaMP6f (*Chen et al., 2013*) was unilaterally injected into the VTA of Gad67-Cre mice (n = 8 mice; *Figure 5a* and *Figure 5—figure supplement 1a*). First, we tested whether GCaMP6f signal from VTA$_{Gad67+}$ neurons correspond to firing frequency in vitro (*Figure 5—figure supplement 1b*). The fluorescence intensity from GCaMP6f was increased in an evoked firing frequency-dependent manner (n = 13 cells; *Figure 5—figure supplement 1c–e*, ΔF/F (%, normalized to 100 Hz), 10 Hz: 9.2 ± 3.0, 20 Hz: 23.0 ± 5.4, 50 Hz: 52.1 ± 6.2). Next, activity recordings were performed in vivo by a fiber optic inserted into the VTA area (*Figure 5—figure supplement 2b*). Offline determination of vigilance states was aided by signals from EEG-EMG electrodes (*Figure 5a–b*). Both fluorescence and EEG-EMG were recorded during the light period in the home cage after habituation. We observed robust changes in the fluorescence signal across brain states (*Figure 5c*, *Figure 5—figure supplement 1f*). To facilitate the statistical analyses of mean ΔF/F among vigilance states, we compared the fluorescence signal at the transition of vigilance states. We found that VTA$_{Gad67+}$ neurons show the highest population activity during NREM and the lowest during REM sleep (*Figure 5d*, *Figure 5—figure supplement 1f*). Notably, VTA$_{Gad67+}$ neurons began to increase their activity before wake-to-NREM transitions (mean ΔF/F: Wake: 2.9 ± 0.4%, NREM: 3.8 ± 0.4%, p=2.5e-6) and decrease their activity before NREM-to-REM (mean ΔF/F: NREM: 5.0 ± 0.5%, REM: 2.7 ± 0.3%, p=2.4e-5) and NREM-to-wake (mean ΔF/F: NREM: 3.8 ± 0.5%, wake: 3.0 ± 0.4%, p=2.4e-4) transitions. However, the changes in signal from REM-to-wake (mean ΔF/F: REM: 2.9 ± 0.4%, wake: 3.7 ± 0.5%, p=0.02) was comparatively less significant and occurred only after the onset of state transition. Most interestingly, the population activity of VTA$_{Gad67+}$ neurons was found to be completely contrary to DA neuronal activity in the VTA (*Dahan et al., 2007*; *Eban-Rothschild et al., 2016*), further

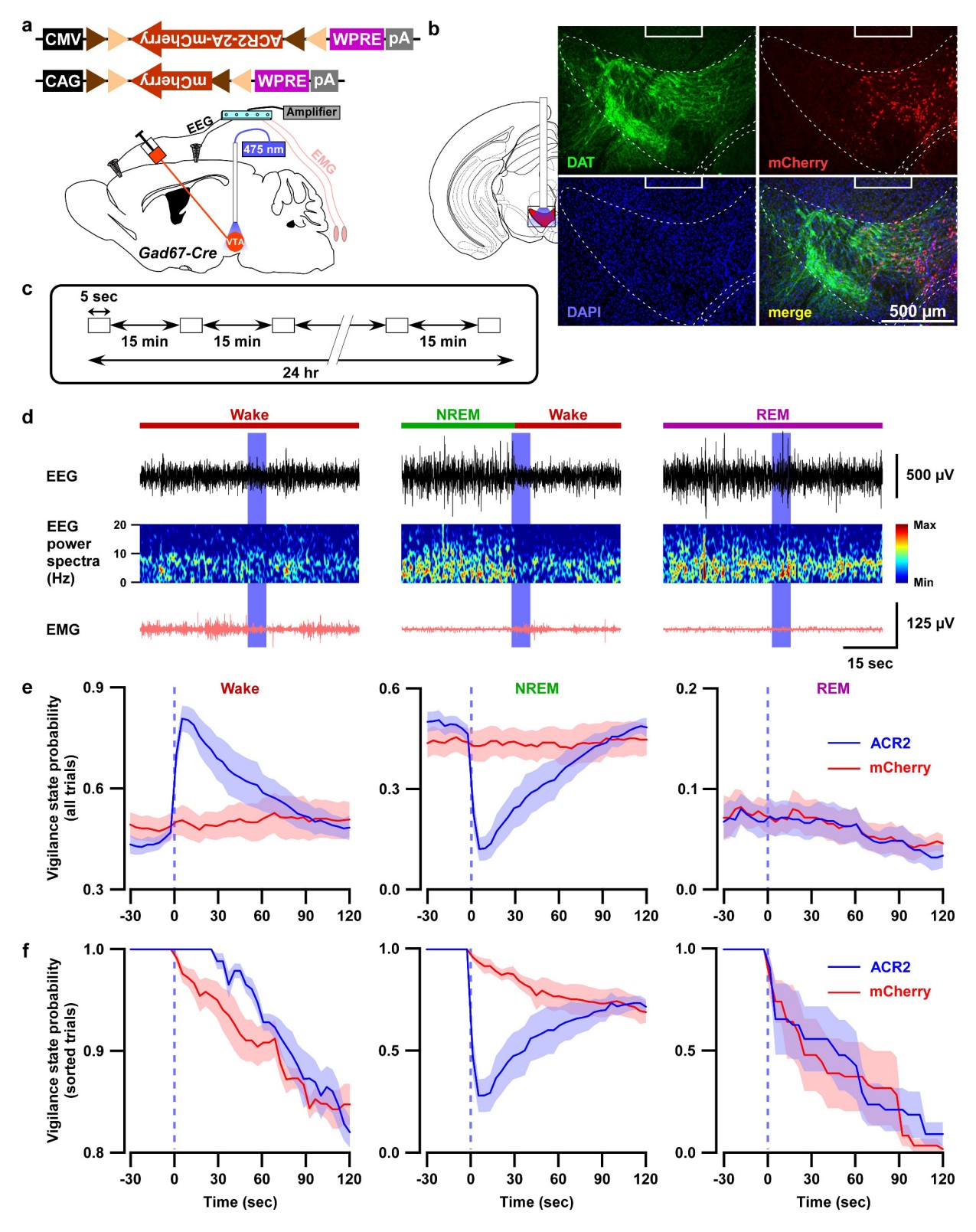

**Figure 3.** Optogenetic inhibition of VTA$_{Gad67+}$ neurons induced wakefulness from NREM sleep, but not from REM sleep. (a) Schematic of surgery showing Gad67-Cre mice expressing either ACR2-2A-mCherry or mCherry alone that were subjected to implantation of fiber optics and EEG-EMG electrodes. (b) Schematic of fiber optic implantation (left). Pictures indicate position of tip of fiber optics and ACR2-2A-mCherry expression and DAT-positive neurons in the VTA. (c) Schematic of protocol for light stimulation in optogenetic inhibition experiments. (d) Representative traces showing

*Figure 3 continued on next page*

Figure 3 continued

EEG, EEG power spectra, and EMG during optogenetic inhibition in different vigilance states (wake, NREM, and REM sleep). Vigilance states are indicated by colored bars above the EEG traces. (e) Probability of vigilance state before and after light illumination in all recorded trials of ACR2-2A-mCherry or mCherry-alone expressing mice. Blue and red lines indicate mean probability of each vigilance state, ACR2-2A-mCherry (n = 6 mice) and mCherry (n = 5 mice). (f) Light illumination during wakefulness, NREM, or REM sleep was isolated from subfigure e (ACR2-2A-mCherry = 6 mice; and mCherry = 5 mice). Each vigilance state lasted for at least 30 s before light illumination was isolated. SEM is indicated as the lighter color band. The online version of this article includes the following source data and figure supplement(s) for figure 3:

**Source data 1.** Source data for *Figure 3e and 3f*.
**Figure supplement 1.** In vitro confirmation of ACR2-mediated optogenetic inhibition of VTA$_{Gad67+}$ neurons.
**Figure supplement 1—source data 1.** Source data for *Figure 3—figure supplement 1f*.

suggesting that Gad67+ neurons and DA neurons differentially modulate sleep-wakefulness in mice (*Eban-Rothschild et al., 2016*).

## VTA$_{Gad67+}$ neurons directly inhibited wake-promoting orexin neurons in the lateral hypothalamus

Dense projections were observed from VTA$_{Gad67+}$ neurons to a well-known sleep-wake regulatory brain region, the lateral hypothalamus (LH), where wake-active and wake-promoting orexin (LH$_{orexin}$) neurons are exclusively located. Thus, we wondered whether VTA$_{Gad67+}$ neurons mediate their sleep-promoting effect through the inhibition of LH$_{orexin}$ neurons. To test this, we generated a bigenic orexin-Flippase (Flp); Gad67-Cre mouse, in which orexin neurons exclusively express Flp recombinase and Gad67+ neurons express Cre recombinase (*Figure 6a–c*) (*Chowdhury et al., 2019*). We injected a Cre-inducible AAV expressing the blue light-gated cation channel channelrhodopsin2 (E123T/T159C; ChR2) (*Berndt et al., 2011*) in the VTA as well as a Flp-inducible AAV expressing tdTomato in the LH of orexin-Flp; Gad67-Cre mice (*Figure 6a–c*). In slice recordings from VTA$_{Gad67+}$ neurons expressing ChR2, blue light flashes (6.8 mW/mm$^2$) through an objective lens could depolarize and significantly increase spontaneous firing frequency to approximately 650% compared with before light illumination (*Figure 6d–f*, n = 5 cells, p=0.004 vs either pre or post, one-way ANOVA followed by post-hoc Tukey). Next, we recorded spontaneous firings from tdTomato-positive neurons (orexin neurons) in the LH by loose cell-attached recordings, and the nerve terminals of VTA$_{Gad67+}$ neurons in the LH were activated by illuminating blue light pulses (*Figure 6g* and *Figure 6—figure supplement 1a*). We found that blue light inhibited LH$_{orexin}$ neuron firing in a light-pulse frequency-dependent manner (5, 10 and 20 Hz). However, no such effect was observed when yellow light pulses (20 Hz) were applied (*Berndt et al., 2011*) (*Figure 6h* and *Figure 6—figure supplement 1b–c*).

To reveal the mechanism of inhibition of orexin neurons, we performed additional electrophysiological experiments. We performed whole-cell voltage clamp recordings from orexin neurons at −60 mV holding potential (mV$_{hold}$) to record post-synaptic currents. Activation of nerve terminals of VTA$_{Gad67+}$ neurons in the LH (blue light pulse, duration of 5 ms) induced a post-synaptic current (PSC) in 8 out of 11 cells. These light-induced PSCs were blocked by gabazine (10 µM), a GABA$_A$ receptor antagonist (*Figure 7a–c*, aCSF: −253 ± 70 pA, gabazine: −9 ± 3 pA, n = 8 cells, p=1.6e-10, paired *t*-test). This result suggests that GABA is involved in generating the light-induced PSCs in LH$_{orexin}$ neurons. The average synaptic delay from light onset was recorded as 6.2 ± 1.0 ms (*Figure 7d*). To rule out the effect of glutamate, we blocked both AMPA (α-amino-3-hydroxy-5-methyl-4-isoxazolepropionic acid) and NMDA (N-Methyl-D-aspartic

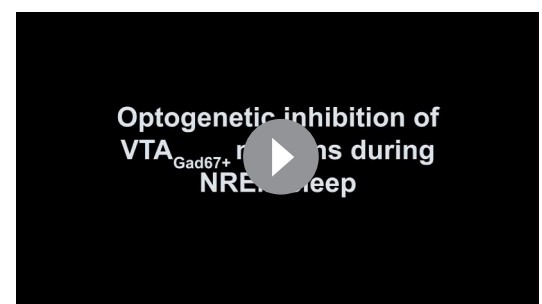

**Video 1.** Optogenetic inhibition of VTA$_{Gad67+}$ neurons during NREM sleep, REM sleep, and wakefulness. Blue light (475 ± 18 nm, 10 mW, 5 s) was illuminated in the VTA area in each vigilance state to inhibit ACR2-expressing VTA$_{Gad67+}$ neurons. EEG, EMG, and light stimulation signals appear at the top of the window. https://elifesciences.org/articles/44928#video1

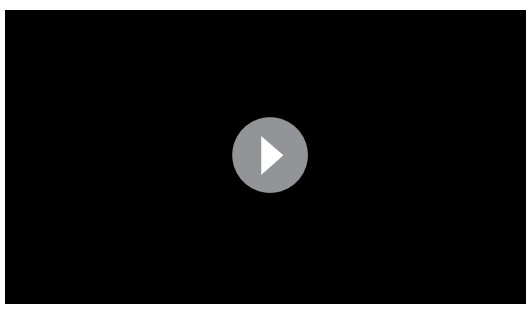

**Video 2.** mCherry control of optogenetic inhibition during NREM sleep, REM sleep, and wakefulness. Blue light (475 ± 18 nm, 10 mW, 5 s) was illuminated in the VTA area in each vigilance state in mCherry-expressing Gad67-Cre mice. EEG, EMG, and light stimulation signals appear at the top of the window.
https://elifesciences.org/articles/44928#video2

acid) type glutamate receptors by applying CNQX (20 µM) and AP5 (50 µM), respectively, in the extracellular bath solution. CNQX and AP5 could not block light-induced PSCs, while the combination of CNQX, AP5, and gabazine could inhibit (*Figure 7e–g*. AP5 +CNQX: −188 ± 38 pA, with gabazine: −6 ± 2 pA, n = 7 cells, p=3.1e-9, paired *t*-test). Again, a delay of 6.7 ± 0.3 ms was found (*Figure 7h*). Finally, to confirm whether light-induced PSCs were indeed driven by monosynaptic release of GABA from VTA$_{Gad67+}$ neurons, we performed an additional set of experiments (*Figure 7i–l*). We found that tetrodotoxin (TTX, 1 µM), a blocker of voltage-gated sodium channels, inhibited the light-induced PSCs (*Figure 7i–k*. aCSF: −340 ± 73 pA, TTX: −2 ± 0.6 pA, n = 6 cells). However, combined application of TTX along with 4-aminopyridine (4-AP, 1 mM), a voltage-gated potassium channel blocker that prolongs depolarization of axon terminals and enables ChR2-mediated release of neurotransmitter in the absence of action potentials (*Petreanu et al., 2009*), could rescue the light-induced PSCs, suggesting a monosynaptic connection between VTA$_{Gad67+}$ neurons and LH$_{orexin}$ neurons (−291 ± 131 pA). Again, the rescued current was blocked by adding gabazine (−6 ± 2 pA), but not by CNQX (−295 ± 121 pA). Finally, to further confirm that Cl$^-$ channels are involved in this GABAergic input, we changed mV$_{hold}$ to +90 mV. The calculated reversal potential of Cl$^-$ under recording conditions were near 0 mV (2.2 mV). As expected, we found that the current direction of light-induced PSCs was opposite at +90 mV$_{hold}$ (*Figure 7j–k*, 237 ± 115 pA). All these experiments confirm that LH$_{orexin}$ neurons were directly innervated and inhibited by VTA$_{Gad67+}$ neurons.

## VTA$_{Gad67+}$ neurons mediate a sleep-promoting function via the lateral hypothalamus

Finally, to test whether the VTA$_{Gad67+}$ to LH projection participates in the induction of NREM sleep, we performed in vivo optogenetic activation of VTA$_{Gad67+}$ nerve terminals in the LH. We expressed either ChR2-EYFP (n = 7 mice) or hrGFP (n = 5 mice) in the VTA$_{Gad67+}$ neurons in a Cre-dependent manner using Gad67-Cre mice. Two weeks later, we bilaterally implanted fiber optics at a diameter of 400 µm above the LH (*Figure 8a and b*). After recovery and habituation, blue light (20 Hz, 10 mW) was applied for 60 min in the dark period when mice were awake and active (*Figure 8c*). Interestingly, blue light illumination into the LH resulted in decreased wakefulness and increased NREM sleep in the ChR2-expressing mice, whereas hrGFP-expressing mice showed no such effects (*Figure 8c–e*, in ChR2-expressing mice: wakefulness (in min) Pre: 55.0 ± 2.2, Stimulation: 29.2 ± 1.7, Post: 53.3 ± 2.1; p=7.3e-8 vs Pre, p=2.7e-7 vs Post, NREM (in min) Pre: 4.3 ± 1.8, Stimulation: 29.9 ± 1.8, Post: 6.0 ± 1.9, p=0 vs Pre, p=5.5e-8 vs Post, one-way ANOVA followed by post-hoc Tukey). Again, the REM sleep remained unaffected (*Figure 8c–e*, in ChR2-expressing mice: REM (in min) Pre: 0.7 ± 0.5, Stimulation: 0.9 ± 0.3, Post: 0.7 ± 0.4, p=0.94 vs Pre, p=0.95 vs Post, one-way ANOVA followed by post-hoc Tukey). These results suggest that VTA$_{Gad67+}$ neurons promote NREM sleep, at least in part, through their projection to the LH.

## Discussion

By employing anterograde tracing and localization of brain-wide neural projections, bidirectional neuronal manipulations, fiber photometry, slice electrophysiology, as well as sleep recordings, we provide multiple lines of evidence in favor of our claim that VTA$_{Gad67+}$ neurons regulate NREM sleep in mice. GABAergic neurons constitute a significant part of the VTA (*Nair-Roberts et al., 2008*; *Pignatelli and Bonci, 2015*) and help to regulate the function of DA neurons residing nearby (*Tan et al., 2012*; *van Zessen et al., 2012*). Dysregulation of signaling pathways in the VTA is associated with drug abuse and several other psychiatric disorders including schizophrenia, bipolar

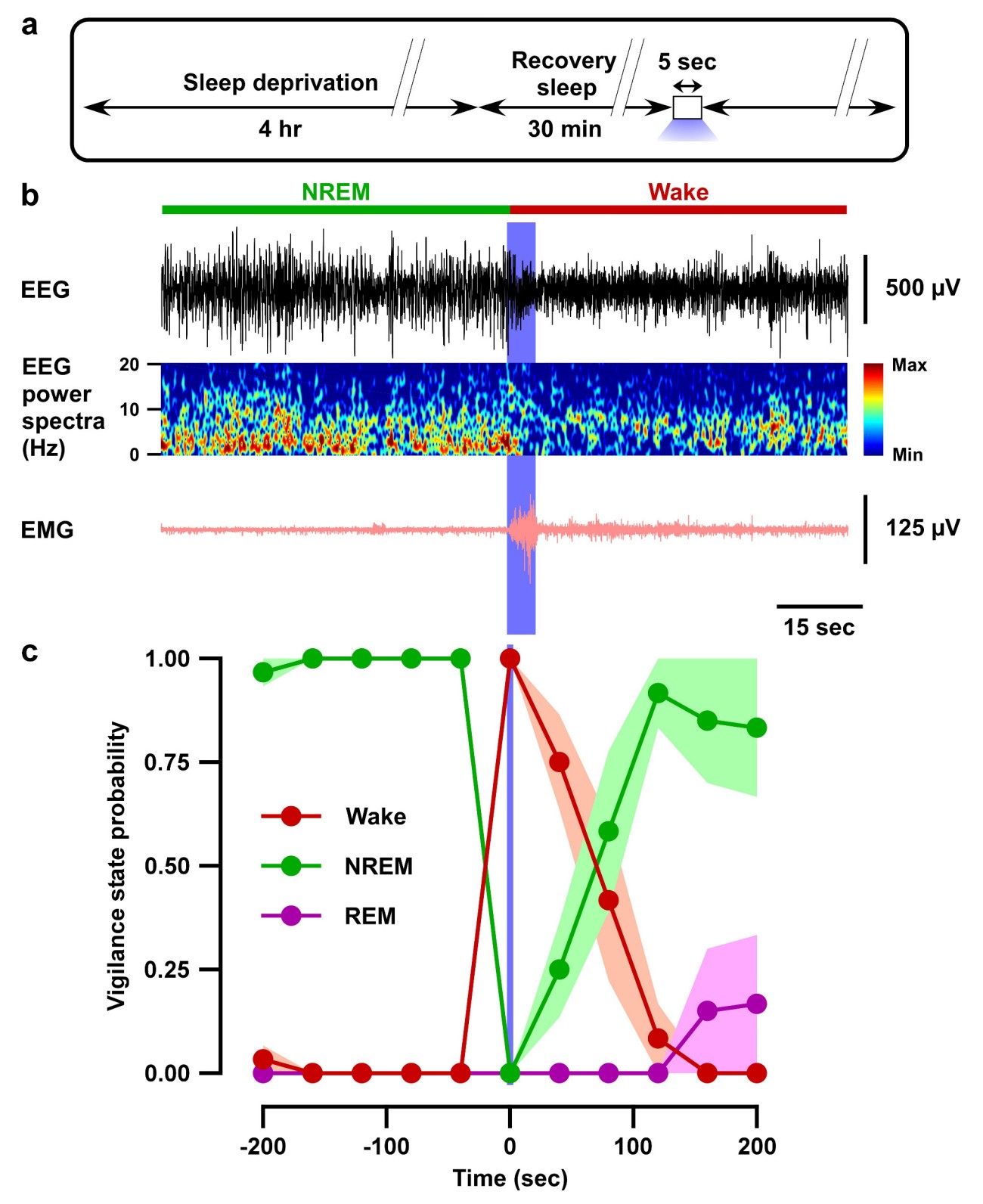

**Figure 4.** Optogenetic inhibition of $VTA_{Gad67+}$ neurons induced wakefulness even under conditions of high homeostatic sleep pressure. (a) Schematic of the protocol of the experiment. (b) EEG, EEG power spectra, and EMG before and after optogenetic inhibition during recovery sleep after 4 hr sleep deprivation. (c) Summary of the experiment in (a) showing the probability of each vigilance state before and after blue light illumination. Colored circles

*Figure 4 continued on next page*

Figure 4 continued

and lines indicate mean probability of each vigilance state and shaded area indicates SEM (ACR2-2A-mCherry: n = 6 mice and mCherry alone: n = 5 mice).

The online version of this article includes the following source data for figure 4:

**Source data 1.** Source data for *Figure 4c*.

disorder, and major depressive disorder (*Winton-Brown et al., 2014*; *Wulff et al., 2010*). Moreover, irregular sleep-wake timing and architectures are recognized as common co-morbidities in many neuropsychiatric and neurodegenerative diseases (*Wulff et al., 2010*). Therefore, the relationship between neurochemical signaling in the VTA and the regulation of sleep/wakefulness poses an interesting point of study. However, classical lesioning experiments suggest that cats with reduced dopamine levels exhibit decreased behavioral arousal but no significant change in electro-cortical waking (*Jones et al., 1973*). It is only very recently that investigators have shown an interest in understanding the role of the VTA in the regulation of sleep/wakefulness (*Eban-Rothschild et al., 2016*; *Oishi et al., 2017a*; *Yang et al., 2018*). However, not much scientific literature has been published focusing on the functional importance of GABAergic neurons in the VTA. Therefore, our findings on the role of these neurons in sleep/wakefulness regulation will provide a conceptual and systematic framework for the association between sleep and psychiatric disorders and will generate opportunities to study VTA-related dysregulation in mental disorders.

van Zassen and colleagues reported that in vivo optogenetic activation of GABAergic neurons in the VTA in mice disrupts reward consummatory behavior (*van Zessen et al., 2012*). In addition, Shank *et al.* reported that dose- and time-related selective ablation of GABAergic neurons in the VTA in rats increased spontaneous locomotor activity (*Shank et al., 2007*). These studies are consistent with the hypothesis that GABAergic neurons in the VTA play an important role in the regulation of behavior. We now argue that one reason for such disruption in behavior might be promotion of NREM sleep by selective activation of GABAergic neurons in the VTA.

Using bidirectional chemogenetic manipulations as well as neurotoxic lesions in rats, a recent study found that neurons in the rostromedial tegmental nucleus (RMTg), also known as the GABAergic tail of the VTA, are essential for physiological NREM sleep (*Yang et al., 2018*). Although Yang *et al.* did not identify neuronal subtypes involved in the RMTg, their results might be related to our findings. Interestingly, $VTA_{Gad67+}$ neurons in our study are located throughout the VTA, but at a somewhat higher density toward the caudal parts of the VTA. More recently, Takata *et al.* reported that GABA neurons in the ventral medial midbrain/pons, which includes the VTA region, regulate sleep/wake cycles by modulating DA neurons (*Takata et al., 2018*). These GABA neurons should include $VTA_{Gad67+}$ neurons. Indeed, GABAergic neurons regulating NREM sleep might be distributed across both VTA and RMTg.

Chemogenetic activation of $VTA_{Gad67+}$ neurons induced NREM sleep accompanied by higher delta power (slow wave) compared with control conditions (*Figure 2g*), suggesting that $VTA_{Gad67+}$ neurons might play a critical role in the generation of slow wave in NREM sleep. Recently, Oishi *et al.* reported that activation of either the cell bodies of GABAergic neurons in the core of NAc or their axonal terminals in the VP evoked slow wave sleep (*Oishi et al., 2017b*). In addition, activation of GABAergic neurons in the basal forebrain, which includes the VP, produced wakefulness, whereas their inhibition induced sleep (*Anaclet et al., 2015*). These facts suggest that inhibition of GABAergic neurons in the VP is a critical pathway to generate slow waves in NREM sleep. We also found that $VTA_{Gad67+}$ neurons moderately project to the VP. Therefore, we reasoned that $VTA_{Gad67+}$ neurons projecting to the VP might be involved in the generation of slow wave sleep.

Population activity recordings across vigilance states shows that DA neurons in the VTA exhibit higher activity in REM sleep versus either wake or NREM sleep (*Eban-Rothschild et al., 2016*). On the contrary, $VTA_{Gad67+}$ neurons exhibit a completely opposite activity pattern from that of DA neurons across vigilance states, with highest activity during NREM sleep (*Figure 5*). This suggests an existing functional interaction between DA and Gad67+ neurons in the VTA. Using in vivo single unit recordings in rats, Lee *et al.* found wake- and REM-active $VTA_{GABA}$ neurons, suggesting that there might be several types of $VTA_{GABA}$ neurons (*Lee et al., 2001*). Our fiber photometry data showed that $VTA_{Gad67+}$ neurons exhibit weak activity even during wakefulness. This might suggest that

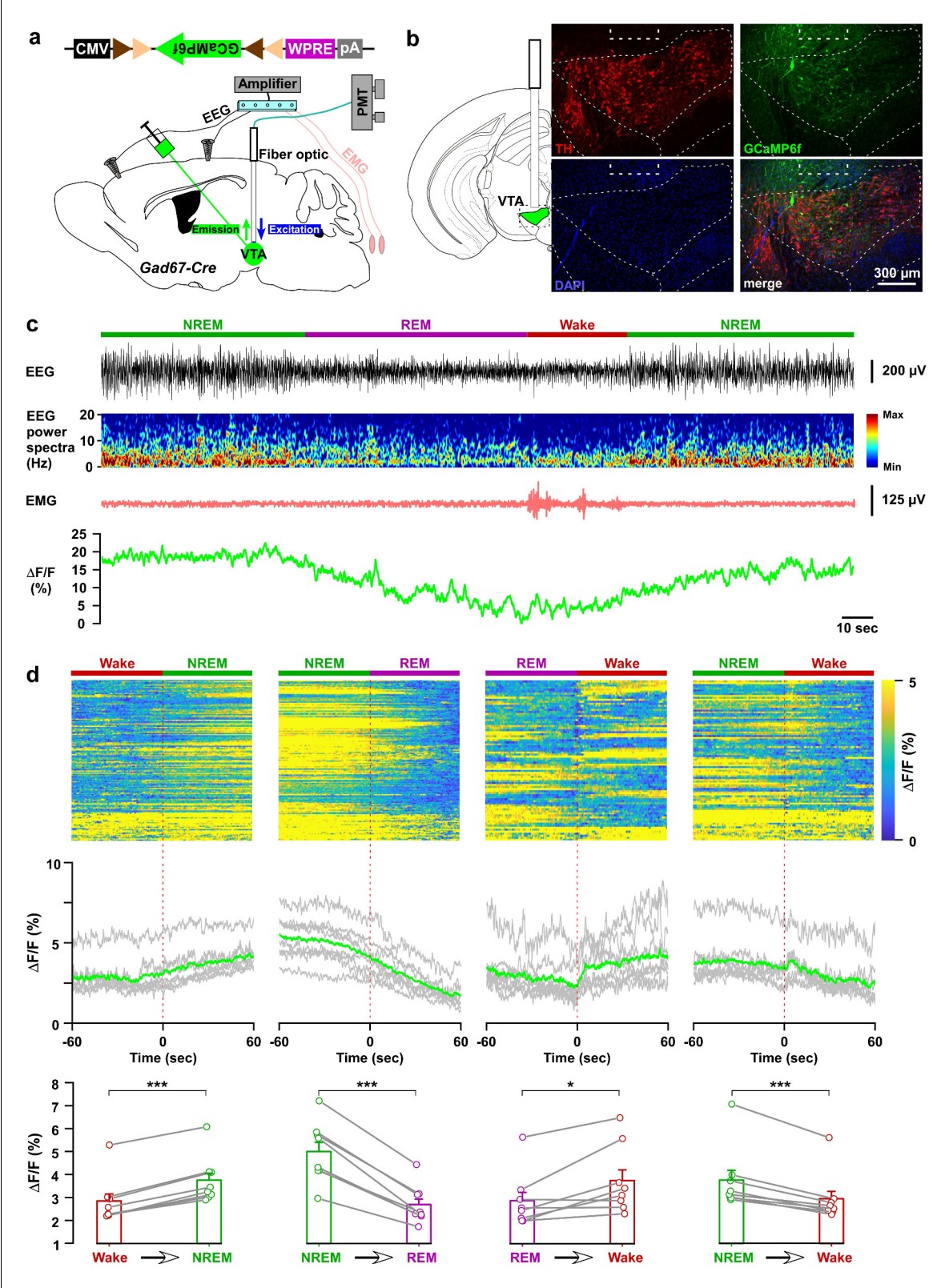

**Figure 5.** In vivo activity recordings of VTA$_{Gad67+}$ neurons using fiber photometry. (a) Schematic of GCaMP6f expression in VTA$_{Gad67+}$ neurons and position of fiber optics. Gad67-Cre mice expressing GCaMP6f were subjected to implantation of guide cannula, EEG, and EMG electrodes. PMT indicates photomultiplier tube. (b) Immunohistochemical studies confirmed that GCaMP6f expression was in the TH-negative cells in the VTA. (c) Representative traces of EEG, EEG spectra, EMG, and fluorescent intensity from GCaMP6f (represented as ΔF/F) in a trial having all different states.

*Figure 5 continued on next page*

*Figure 5 continued*

Vigilance states were determined by EEG and EMG signals and indicated by colored bars. (**d**) Fluorescent intensity alterations in each trial 60 s before and after vigilance state changes. Upper panel shows the heat map of all separated transitions. Middle panel represents the changes in the intensity of calcium signals represented as ΔF/F. Gray lines indicate average intensities in individual mice and the green line indicates the mean of all mice. Lower panel indicates the average intensity separated for specific vigilance states. Data are represented as mean ± SEM. ***, $p<0.001$; *, $p<0.05$. Two-tailed paired Student's *t*-test (n = 8 mice).

The online version of this article includes the following source data and figure supplement(s) for figure 5:

**Source data 1.** Source data for *Figure 5d*.
**Figure supplement 1.** GCaMP6f-mediated recordings of VTA$_{Gad67+}$ neuronal activity both in vitro and in vivo.
**Figure supplement 1—source data 1.** Source data for *Figure 5—figure supplement 1d and e*.
**Figure supplement 2.** Histological verification of tip of fiber optics in mice used in behavioral experiments.

VTA$_{Gad67+}$ neurons are also comprised of several subtypes. However, most VTA$_{Gad67+}$ neurons are predominantly active in NREM sleep. Therefore, additional research is needed to clarify any electrophysiological, anatomical, and/or functional variations of GABAergic neurons in the VTA. Very recently, Yu *et al.* reported that coordinated interaction between GABA and glutamate neurons in the VTA regulates sleep/wakefulness in mice (*Yu et al., 2019*). Although this study similarly found a NREM-sleep promoting role of VTA$_{GABA}$ neurons, the in vivo activity of VTA$_{GABA}$ neurons was quite different from our observations with the highest activity observed during wake and REM sleep. One difference between studies was that Yu *et al.* used Vgat-Cre mice (Cre recombinase is targeted to the *slc32a1* gene) and we used Gad67-Cre mice to target GABAergic neurons. Our in situ hybridization results showed that Gad67-positive neurons represent a small subset of Vgat-positive neurons in the VTA. Again, this difference further indicates the existence of different populations of GABAergic neurons in the VTA.

Optogenetic inhibition of VTA$_{Gad67+}$ neurons induced immediate wakefulness from NREM sleep, but not from REM sleep, suggesting the importance of uninterrupted neuronal activity of VTA$_{Gad67+}$ neurons for the maintenance of NREM sleep. Both chemogenetic activation and optogenetic inhibition data suggest that VTA$_{Gad67+}$ neurons might not play a decisive role in the physiological regulation of REM sleep. Interestingly, inhibition of VTA$_{Gad67+}$ neurons prolonged wakefulness (*Figure 3f*). This result might suggest that VTA$_{Gad67+}$ neurons also regulate levels of wakefulness. This is consistent with data showing that VTA$_{Gad67+}$ neurons displayed weak activity in wakefulness in terms of population activity. This idea is also supported by observed increases in spontaneous locomotor activity following selective ablation of VTA$_{GABA}$ neurons in rats (*Shank et al., 2007*). These facts might suggest that an improvement in alertness and ability to maintain wakefulness require the suppression of activity of VTA$_{GABA}$ neurons.

One possible cellular mechanism underlying NREM sleep promotion by VTA$_{GABA}$ neurons is via inhibition of DA neurons residing in the VTA. In addition to this, our results showed that direct inhibition of wake-promoting LH$_{orexin}$ neurons might contribute to the induction of NREM sleep. Projection-specific activation of VTA$_{GABA}$ neuron nerve terminals using optogenetics indicated that VTA$_{GABA}$-to-LH represents a pathway responsible for inducing NREM sleep. It will also be fascinating to study how VTA$_{GABA}$ neurons are regulated. Using an optimized trans-synaptic retrograde tracing approach, Faget and colleagues recently labeled afferent neurons to DA, GABA, or glutamate neurons in the VTA and found that these populations receive qualitatively similar inputs, with dominant and comparable projections from three brain areas known to be critical for sleep/wakefulness regulation: LH, raphe, and ventral pallidum (*Faget et al., 2016*). Here we report that VTA$_{Gad67+}$ neurons project to those areas, suggesting the existence of a mutual interaction with these brain areas to regulate sleep/wakefulness. Moreover, while many brain areas including the cholinergic basal forebrain and brain stem, histaminergic posterior hypothalamus, serotonergic raphe nucleus, as well as the noradrenergic locus coeruleus are important in sleep/wake regulation (*Brown et al., 2012*), we speculate a critical role for VTA$_{Gad67+}$ neurons in sleep/wakefulness-regulating circuitry as these neurons modulate diverse targets, including DA neurons in the VTA (*Tan et al., 2012*).

In conclusion, our study elucidated that VTA$_{GABA}$ neurons regulate NREM sleep in mice. These neurons might be a possible target for therapeutic intervention in treating sleep-related as well as neuropsychiatric disorders.

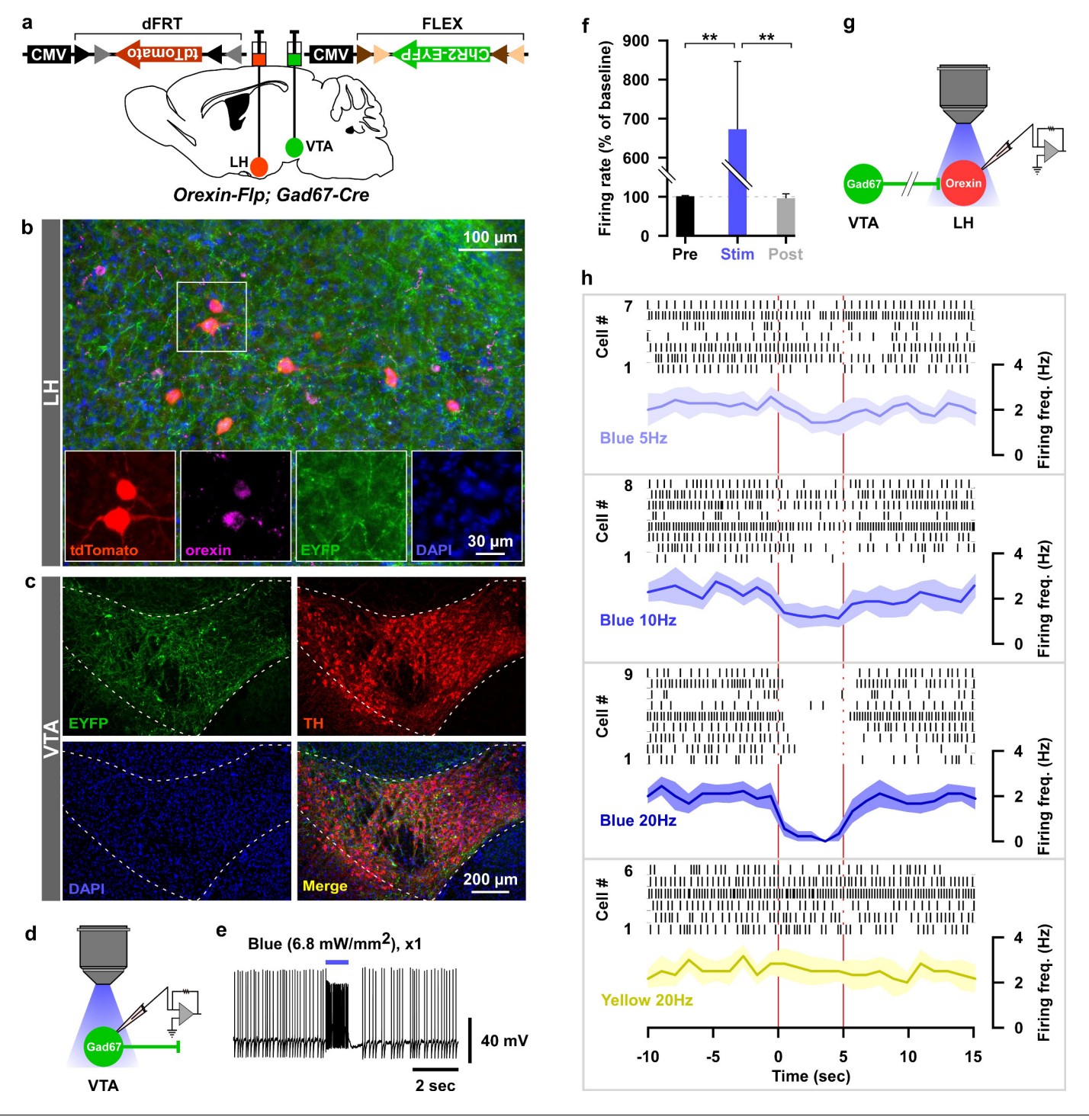

**Figure 6.** Optogenetic activation of VTA$_{Gad67+}$ neuronal terminals in the LH inhibited orexin neurons in vitro. (**a**) AAV-mediated gene expression in orexin-Flp; Gad67-Cre bigenic mice. (**b and c**) Immunohistochemical studies confirmed expression of tdTomato exclusively in orexin neurons and ChR2 in non-TH-positive neurons in the VTA. (**d–f**) Schematic and current clamp recordings from ChR2-expressing Gad67+ neurons in the VTA in acute brain slices. Blue light stimulation of 6.8 mW/mm$^2$ increased the firing up to 674 ± 174% (n = 5 cells, p=0.004 vs both pre and post, one-way ANOVA followed by Tukey post hoc tests). (**g**) Schematic of the experiments in (**h**). (**h**) Firing of LH$_{orexin}$ neurons in vitro and effect of activation of VTA$_{Gad67+}$ neuronal terminals using different frequencies of blue lights (5, 10, or 20 Hz). Yellow light of 20 Hz was used as a negative control. Raster plot of each trial (upper panel) and running average of firing frequencies (lower panel) of LH$_{orexin}$ neurons are indicated in each rectangular box following illumination of the brain slice through the objective lens. Two vertical red lines indicate illumination start (left) and stop (right) timing.

*Figure 6 continued on next page*

*Figure 6 continued*

The online version of this article includes the following source data and figure supplement(s) for figure 6:

**Source data 1.** Source data for *Figure 6f*.
**Figure supplement 1.** Optogenetic activation of nerve terminals of VTA$_{Gad67+}$ neurons in the LH inhibited orexin neurons in a blue-light pulse frequency-dependent manner.
**Figure supplement 1—source data 1.** Source data for *Figure 6—figure supplement 1c*.

# Materials and methods

## Key resources table

| Reagent type (species) or resource | Designation | Source or reference | Identifiers | Additional information |
|---|---|---|---|---|
| Strain, strain background | AAV(9)-CMV-FLEX-hrGFP | This paper | NA | Titer: $6.0 \times 10^{12}$ copies/ml |
| Strain, strain background | AAV(9)-CAG-FLEX-hM3Dq-mCherry | This paper | NA | Titer: $1.1 \times 10^{12}$ copies/ml |
| Strain, strain background | AAV(9)-CMV-FLEX-ACR2-2A-mCherry | *Mohammad et al., 2017* | Genbank accession# KP171709 | Titer: $6.2 \times 10^{12}$ copies/ml |
| Strain, strain background | AAV(9)-CAG-FLEX-mCherry | This paper | NA | Titer: $1.9 \times 10^{12}$ copies/ml |
| Strain, strain background | AAV(9)-CMV-FLEX-ChR2 (ET/TC)-eYFP | *Berndt et al., 2011* | NA | Titer: $3.0 \times 10^{13}$ copies/ml |
| Strain, strain background | AAV(9)-CMV-FLEX-GCaMP6f | *Chen et al., 2013* | NA | Titer: $1.3 \times 10^{12}$ copies/ml |
| Strain, strain background | AAV(DJ)-CMV-dFrt-tdTomato-WPRE | This paper | NA | Titer: $8.1 \times 10^{12}$ copies/ml |
| Genetic reagent (*Mus musculus*) | Glutamic acid decarboxylase 67-Cre | *Higo et al., 2009* | Gad67-Cre | |
| Genetic reagent (*Mus musculus*) | Orexin-Flippase | *Chowdhury et al., 2019* | Orexin-Flp | |
| Genetic reagent (*Mus musculus*) | Orexin-Flippase; Glutamic acid decarboxylase 67-Cre | *Chowdhury et al., 2019* | Orexin-Flp; Gad67-Cre | |
| Antibody | Rabbit polyclonal anti-TH | Millipore | AB-152 | (1/1000) |
| Antibody | Mouse monoclonal anti-GFP | Fujifilm Wako Pure Chemical Corporation | mFX75 | (1/1000) |
| Antibody | Goat polyclonal anti-orexin | Santa Cruz Biotechnology | sc-8070 | (1/1000) |
| Antibody | Mouse monoclonal anti-Gad67 | Millipore | MAB5406 | (1/500) |
| Antibody | Mouse monoclonal anti-DAT | Frontier Institute Co. Ltd. | DAT-Rb-Af1800 | (1/1000) |
| Antibody | Mouse monoclonal anti-DsRED | Santa Cruz Biotechnology | sc-390909 | (1/1000) |
| Commercial assay or kit | RNAscope Fluorescent Multiplex Reagent ver.2 | ACD Bio | | |
| Chemical compound, drug | clozapine-N-oxide (CNO) | Enzo Life Sciences | BML-NS105-0025 | |

*Continued on next page*

*Continued*

| Reagent type (species) or resource | Designation | Source or reference | Identifiers | Additional information |
|---|---|---|---|---|
| Chemical compound, drug | 4',6-diamidino-2-phenylindole dihydrochloride (DAPI) | Thermo Fisher Scientific | Cat# D1306 | |
| Chemical compound, drug | Gabazine | Abcam | Cat# Ab-120042 | |
| Chemical compound, drug | tetrodotoxin (TTX) | Alomone Labs | Cat# T-550 | |
| Chemical compound, drug | D-2-Amino-5-phosphopentanoic acid (AP5) | Alomone Labs | Cat# D-145 | |
| Chemical compound, drug | 6-cyano-7-nitroquinoxaline-2,3-dione (CNQX) | Sigma-Aldrich | Cat# C127 | |
| Chemical compound, drug | 4-Aminopyridine (4-AP) | Sigma-Aldrich | Cat# 8.01111 | |
| Software, algorithm | Origin 2017 | Lightstone | Origin 2018 | |
| Software, algorithm | SleepSign | Kissei Comtec | Version 3 | |
| Software, algorithm | pClamp 10.5 Software and Algorithms | Molecular Devices | RRID:SCR_011323 | |
| Software, algorithm | ImageJ | https://imagej.nih.gov/ij/ | RRID:SCR_003070 | |
| In situ hybridyzation probe | Gad67 Channel 1 | ACD Bio | Cat# 400951 | |
| In situ hybridyzation probe | Vgat Channel 4 | ACD Bio | Cat# 319191 | |
| In situ hybridyzation probe | mCherry Channel 3 | ACD Bio | Cat# 431201 | |
| TSA enhancement (ISH) | Opal520 | PerkinElmer | | |
| TSA enhancement (ISH) | Opal620 | PerkinElmer | | |
| TSA enhancement (ISH) | Opal690 | PerkinElmer | | |

## Animals

All experimental protocols that involved animals were approved by the Institutional Animal Care and Use Committees, Research Institute of Environmental Medicine, Nagoya University, Japan (Approval number #18232, #18239). All efforts were made to reduce the number of animals used and also to minimize the suffering and pain of animals. Animals were maintained on a 12 hr light-dark cycle (lights were turned on at 8:00 am), with free access to food and water.

## Generation and microinjection of adeno-associated virus (AAV) vectors

AAV vectors were produced using the AAV Helper-Free System (Agilent Technologies, Inc, Santa Clara, CA). The virus purification method was adopted from a previously published protocol (*Inutsuka et al., 2016*). Briefly, HEK293 cells were transfected together with three distinct plasmids carrying a pAAV vector, pHelper and pAAV-RC (serotype 9 or DJ; purchased from Cell Biolabs Inc, San Diego, CA) using a standard calcium phosphate method. HEK293 cells were collected and suspended in artificial CSF (aCSF) solution (in mM: 124 NaCl, 3 KCl, 26 NaHCO$_3$, 2 CaCl$_2$, 1 MgSO$_4$, 1.25 KH$_2$PO$_4$ and 10 glucose) three days post-transfection. Following multiple freeze-thaw cycles,

the cell lysates were treated with benzonase nuclease (Merck, Darmstadt, Germany) at 37°C for 30 min, and were centrifuged 2 times at 16,000 g for 10 min at 4°C. The supernatant was used as the virus-containing solution. Quantitative PCR was performed to measure the titer of purified virus. Virus aliquots were then stored at −80°C until use.

Adult Gad67-Cre or orexin-Flp; Gad67-Cre mice of both sexes were subjected to either unilateral or bilateral injection of AAV(9)-CMV-FLEX-hrGFP (100 × 1 nl, 6.0 × $10^{12}$ copies/ml), AAV(9)-CAG-FLEX-hM3Dq-mCherry (200 × 2 nl, 1.1 × $10^{12}$ copies/ml), AAV(9)-CMV-FLEX-ACR2-2A-mCherry (300 × 2 nl, 6.2 × $10^{12}$ copies/ml), AAV(9)-CAG-FLEX-mCherry (300 × 2 nl, 1.9 × $10^{12}$ copies/ml), AAV(9)-CMV-FLEX-ChR2 (ET/TC)-eYFP (300 nl, 3.0 × $10^{13}$ copies/ml), or AAV(9)-CMV-FLEX-GCaMP6f (300 × 1 nl, 1.3 × $10^{12}$ copies/ml) into the VTA (3.0 to 3.7 mm posterior and 0.4 to 0.6 mm lateral from bregma, 4.0 to 4.2 mm deep from brain surface) under ~1.2% isoflurane (Fujifilm Wako Pure Chemical Industries, Osaka, Japan) anesthesia. Orexin-Flp; Gad67-Cre bigenic mice also received bilateral injection of AAV(DJ)-CMV-dFrt-tdTomato-WPRE (300 × 2 nl, 8.1 × $10^{12}$ copies/ml) into the lateral hypothalamus (1.5 mm posterior and 0.5 mm lateral from bregma, 5.0 mm deep from brain surface), which were used for slice electrophysiological experiments.

## Immunohistochemistry

Under deep anesthesia with 0.65% pentobarbital sodium solution (Kyoritsu Seiyaku Corporation, Tokyo, Japan) diluted with saline (1.0 ml/kg body weight), mice were subjected to serial transcardial perfusion first using ice-cold saline (20 ml) and then ice-cold 4% formaldehyde solution (20 ml, Fujifilm Wako Pure Chemical Industries, Ltd., Osaka, Japan). The brain was then gently collected and post-fixed with 4% formaldehyde solution at 4°C overnight. Later, the brain was subsequently immersed in phosphate-buffered saline (PBS) containing 30% sucrose at 4°C for at least 2 days. Coronal sections of either 40 or 80 µm thickness were made using a cryostat (Leica CM3050 S; Leica Microsystems, Wetzlar, Germany; or Leica VT1000 S, Wetzlar, Germany), and slices were preserved in PBS containing 0.02% of $NaN_3$ at 4°C until stained. For staining, coronal brain sections were immersed in blocking buffer (1% BSA and 0.25% Triton-X in PBS), and then incubated with primary antibodies (TH: Millipore, Massachusetts, 1/1000 dilution; DAT: Frontier Institute Co. Ltd., Hokkaido, Japan, 1/1000 dilution, Japan; DsRED: Santa Cruz Biotechnology, Heidelberg, Germany, 1/1000 dilution; GFP: Fujifilm Wako Pure Chemical Corporation, Osaka, Japan, 1/1000 dilution; orexin-A: Santa Cruz Biotechnology, 1/1000 dilution) at 4°C overnight. For Gad67 staining, slices were incubated with anti-Gad67 antibody (Millipore, 1/500 dilution in blocking buffer without Triton-X) at 4°C for 4 days. After washing by blocking buffer three times, the brain sections were then incubated with secondary antibodies for 1 hr at room temperature. After washing with the same blocking solution three times, slices were stained by DAPI (Thermo Fisher Scientific, Waltham, MA) across several experiments. Slices were mounted in 50% glycerol solution and examined with an epifluorescence microscope (BZ-9000, Keyence, Osaka, Japan or IX71, Olympus, Tokyo, Japan).

## Anterograde tracing and localization of brain-wide neural projections

A Cre-inducible AAV carrying the hrGFP gene (AAV(9)-CMV-FLEX-hrGFP; 100 × 1 nl, 6.0 × $10^{12}$ copies/ml) was unilaterally injected into the VTA of Gad67-Cre mice. Three weeks post-injection, animals were perfused-fixed and brain slices of 80 µm thickness were made serially from the anterior to the posterior part of the brain using a vibratome (Leica VT1000 S, Wetzlar, Germany). After DAPI staining, slices were serially mounted and images were taken using an epifluorescence microscope (BZ-9000, Keyence, Osaka, Japan or IX71, Olympus, Tokyo, Japan). Images were taken using an identical configuration in the microscope and were then analyzed using ImageJ (US National Institute of Health) software. Projection scorings were made in all visible projection sites, except for the VTA, by first selecting the most innervated brain region and comparing other areas to that region.

## In situ RNA hybridization using RNAscope

Under deep anesthesia with 0.65% pentobarbital sodium solution (Kyoritsu Seiyaku Corporation, Tokyo, Japan) diluted with saline (1.0 ml/kg body weight), mice were transcardially perfused using ice-cold saline (20 ml) and then using ice-cold 4% formaldehyde solution (20 ml, Fujifilm Wako Pure Chemical Industries, Ltd., Osaka, Japan). The brain was then gently removed, post-fixed with 4% paraformaldehyde solution overnight at 4°C and was subsequently immersed in phosphate-buffer

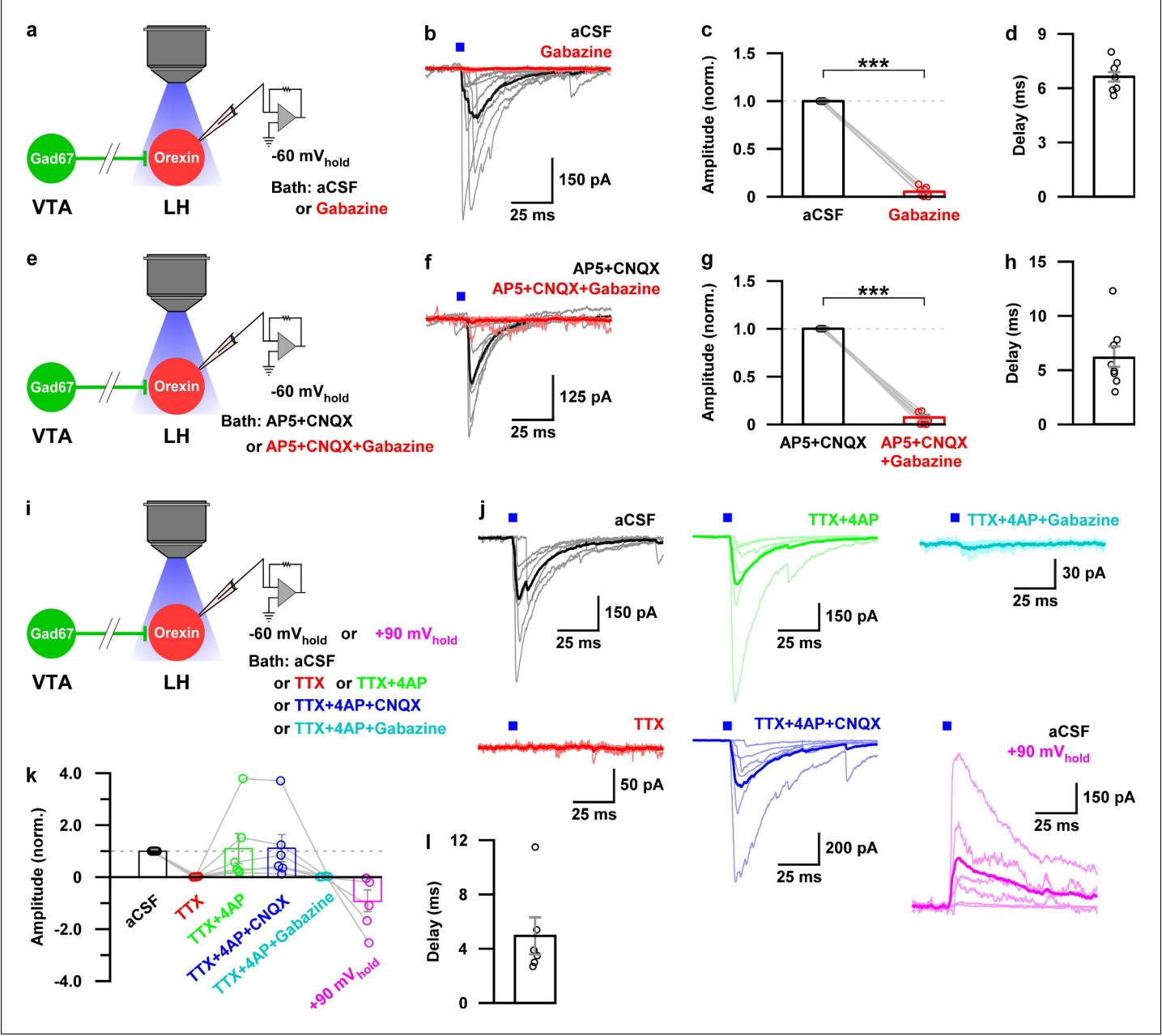

**Figure 7.** Monosynaptic GABAergic input underlies the inhibitory effect of VTA$_{Gad67+}$ neurons onto LH$_{orexin}$ neurons. (a) Schematic of experiments in b-d. (b) Blue light pulses (5 ms) induced post-synaptic currents in the LH$_{orexin}$ neurons. The thicker line indicates average traces, and the thinner line indicates responses in individual cells (n = 8 cells). (c) Summary of the experiments in b showing the amplitude of current normalized to the aCSF application. (d) Delay in response from light onset. (e) Schematic of the experiments in f-h. (f–h) Similar data representation as in (b–e) in the presence of glutamatergic and GABAergic antagonists. (i) Schematic of the experiments j-l. (j) The effect of glutamatergic and GABAergic antagonists and channel blockers on blue light pulse-induced currents (n = 7 cells). (k) Summary of the experiments in (j) showing the amplitude of current normalized to aCSF (n = 6 cells). Data are represented as mean ± SEM. ***, p<0.001. p values were calculated by two-tailed paired Student's t-test. The online version of this article includes the following source data for figure 7:

**Source data 1.** Source data for *Figure 7c, 7g and 7k*.

containing 30% sucrose at 4°C for at least 2 days. Coronal brain sections of 25 μm thickness were made using a cryostat (Leica CM3050 S; Leica Microsystems) and were mounted on glass slides (SMAS-01, Matsunami, Japan), and fixed in 4% paraformaldehyde for 60 min. The slides were then treated by RNAscope multiplex fluorescent v2 (#323100, Advanced Cell Diagnostics, Hayward, CA)

according to the RNAscope standard protocol. In short, slides were incubated with hydrogen peroxide at RT for 10 min, followed by boiling with target retrieval reagent at 98 ~ 102°C for 5 min, and protease digestion at 40°C in a HybEZ hybridization oven (Advanced Cell Diagnostics) for 30 min. Subsequently, slides were incubated at 40°C with target probes in hybridization buffer for 2 hr, AMP1 in hybridization buffer for 30 min, AMP2 in hybridization buffer for 30 min, and AMP3 in hybridization buffer for 15 min. After each hybridization step, slides were washed with wash buffer twice at room temperature. For multiplex detection, equimolar amounts of target probes, AMP1, AMP2 and AMP3 of each amplification system were used. Sequences of target probes, AMP1, AMP2, and AMP3 are proprietary (vGAT, GAD1 and mCherry, Advanced Cell Diagnostics, Hayward, CA). For fluorescent detection, the RNA probes were conjugated to Opal 520, 620, or 690 with the HRP and TSA Plus fluorophores system (Perkin Elmer, Waltham, MA). Slices were mounted with Pro-Long Gold Antifade Mountant (Thermo Fisher Scientific, Waltham, MA) and observed via a confocal microscope (LSM 710 Zeiss, Oberkochen, Germany).

## Surgery for EEG-EMG and/or optogenetics, fiber photometry

Procedures for implanting EEG and EMG electrodes for polysomnographic recording experiments were adapted from the previously published protocol (*Tabuchi et al., 2014*). Briefly, virus-injected mice were implanted with EEG and EMG electrodes under isoflurane anesthesia. Immediately after surgery, each mouse received an i.p. injection of 10 ml/kg of an analgesic solution containing 0.5 mg/ml of Carprofen (Zoetis Inc, Parsippany-Troy Hills, NJ). Mice were singly housed for 7 days during the recovery period. Mice were then connected to a cable in order to allow them to move freely in the cage as well as to be habituated to the recording cable for another 7 days.

For fiber-guided optogenetic experiments, virus-injected mice received a surgical implantation of single fiber optic cannula (400 μm; Lucir Inc, Japan), along with EEG-EMG electrodes, above the VTA (AP −3.3 to −3.6 mm; ML 0.4 to 0.6 mm; DV −3.75 mm). For fiber photometry experiments, virus-injected mice received surgical implantation of a single guide cannula (400 μm; Thorlabs Inc) just above the VTA (AP −3.3 mm; ML 0.4 to 0.5 mm; DV −4.0 mm) to target $VTA_{Gad67+}$ neurons. These mice were also implanted with the EEG-EMG electrodes following the protocol described above.

For projection-specific optogenetic activation of $VTA_{Gad67+}$ nerve terminals in the LH, virus-injected mice received bilateral implantation of fiber optic cannula (400 μm; Kyocera Corporation, Japan), along with EEG-EMG electrodes, above the LH at a stereotaxic co-ordinate of AP −1.4 mm; ML ± 0.9 mm; DV −5.0 mm.

## Vigilance state determination

EEG and EMG signals were amplified (AB-610J, Nihon Koden, Japan), filtered (EEG at 1.5–30 Hz, and EMG at 15–300 Hz), digitized (at a sampling rate of 128 Hz), and recorded (Vital Recorder, Kissei Comtec Co., Ltd, Japan) from individual habituated mice. Recorded signals were then analyzed to identify vigilance states using SleepSign (Kissei Comtec) software. Vigilance state identification was assisted by an infrared sensor as well as by video monitoring through a CCD video camera (Amaki Electric Co., Ltd., Japan) during both the light and dark periods (Kissei Comtec). Video recording during the dark period was aided by infrared photography (Amaki Electric Co., Ltd., Japan). EEG and EMG data were automatically scored in epochs (every 4 s) and classified as wake, REM sleep, or NREM sleep. All auto-screened data were examined visually and corrected. The EEG analysis yielded power spectra profiles over a 0 ~ 20 Hz window with 1 Hz resolution for delta (1–5 Hz), theta (6–10 Hz), alpha (11–15 Hz), and beta (16–20 Hz) bandwidths. The criteria for determining vigilance states were the same as the protocol described elsewhere (*Tabuchi et al., 2014*): briefly, (i) wake (low EEG amplitude with high EMG or locomotion score), (ii) NREM sleep (low EMG and high EEG delta amplitude), and (iii) REM sleep (low EMG as well as low EEG amplitude with high theta activity, and should be followed by NREM).

## In vivo recordings and data analysis of neuronal activity using fiber photometry

In vivo population activity of the $VTA_{Gad67+}$ neurons was recorded using a silica fiber of 400 μm by implanting the fiber just above the VTA. Details of the fiber photometric recordings are described

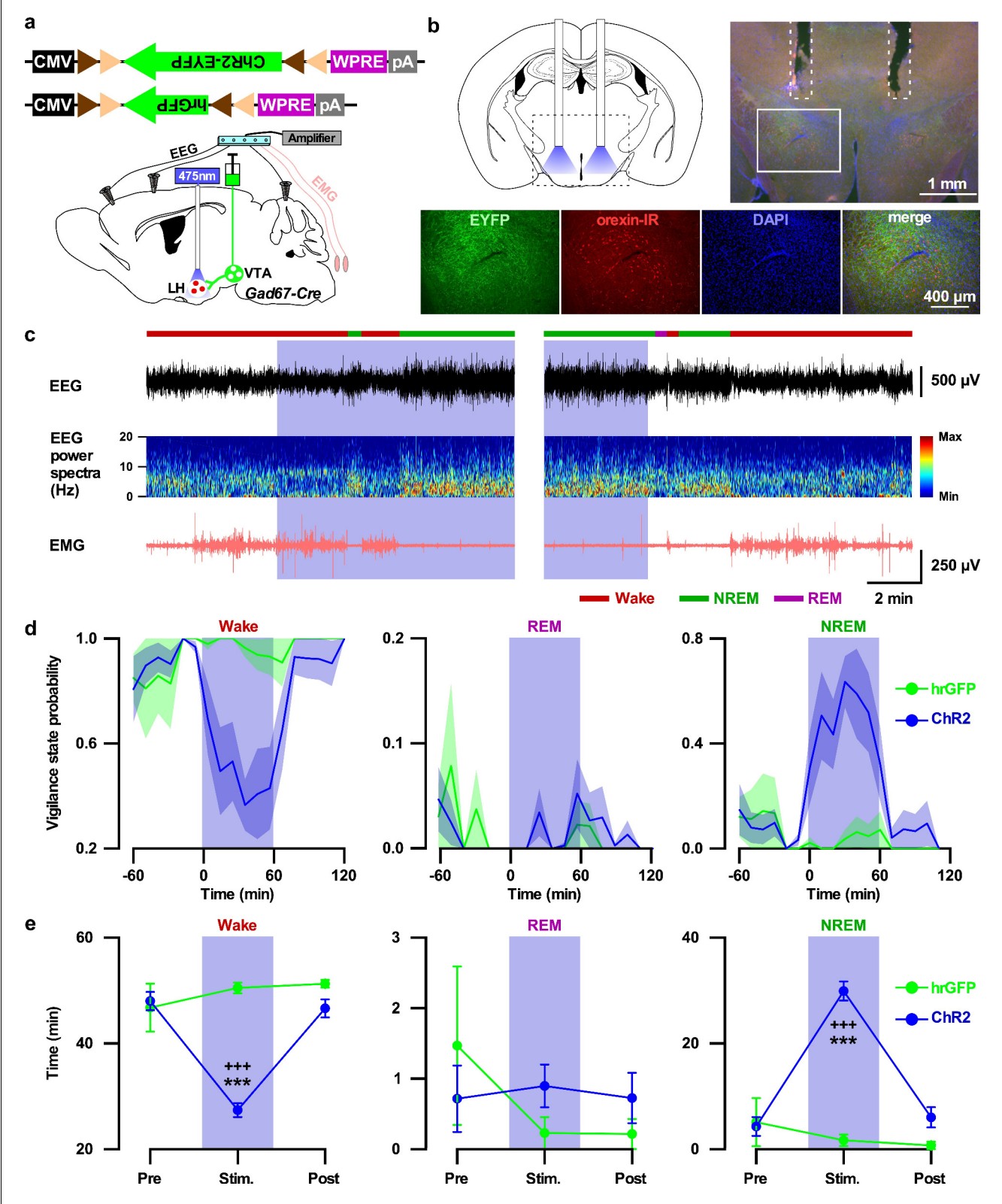

**Figure 8.** Activation of VTA<sub>Gad67+</sub> nerve terminals in the LH promoted NREM sleep. (a) Schematic of the projection specific optogenetic activation. (b) Schematic and immunohistochemistry of fiber implantation in the LH. Top right panel shows merged fluorescence image depicted in the coronal brain map. Lower panels indicate enlarged images of the boxed area shows VTA<sub>Gad67+</sub> terminals near orexin neurons in the LH. (c) Representative traces showing EEG, EEG spectra, and EMG while mice experienced 20 Hz blue light stimulation (left: start of light illumination, right: stop of light

*Figure 8 continued on next page*

*Figure 8 continued*

illumination). Vigilance states determined by EEG and EMG signals are indicated by colored bars. (**d** and **e**) Probability of vigilance state (**d**) and total time spent (**e**) of either ChR2- or hrGFP-expressing mice. Blue and green lines indicate mean probability of each vigilance state in ChR2 (n = 7 mice) and hrGFP (n = 5 mice) group, respectively. Data are represented as mean ± SEM. ***, p<0.001 (vs Pre); +++, p<0.001 (vs Post). p values were calculated by one-way ANOVA followed by post-hoc Tukey test.

The online version of this article includes the following source data for figure 8:

**Source data 1.** Source data for *Figure 8d and 8e*.

elsewhere (*Inutsuka et al., 2016*). Briefly, the fiber photometry system (COME2-FTR/OPT, Lucir, Tsukuba, Japan) utilizes a custom-made single silica fiber of 400 μm diameter to deliver excitation light and to detect fluorescence from GCaMP6f, simultaneously. Blue excitation light (465 nm, 0.5 mW at the tip of the silica fiber) was produced by a high-power LED system (PlexonBright OPT/LED Blue_TT_FC, Plexon, Dallas, TX). The LED-emitted excitation light was reflected by a dichroic mirror and coupled to the silica fiber (400 μm diameter) through an excitation bandpass filter (path 472 ± 35 nm). GCaMP6f-emitted green fluorescence was collected by the same silica fiber passed through a bandpass emission filter (path 525 ± 25 nm) and guided to a photomultiplier (PMTH-S1M1-CR131, Zolix instruments, Beijing, China). The fiber photometry signal was recorded by Vital Recorder (Kissei Comtec Co., Ltd, Japan) along with the EEG/EMG signals. Fiber photometry signals were collected at a sampling frequency of 128 Hz and the software averaged every 10 samples to minimize fluctuations and noise.

After recording and sleep analysis, the fiber photometry signal was outputted along with the EEG and EMG signals as a text file of raw data. For each experiment, the photometry signals at all data points were motion averaged and were then converted to ΔF/F by $\Delta F/F(t) = (F(t) - F_{min})/F_{min}$. We recorded the signals in the light period as nocturnal animal mice usually show multiple transitions among different vigilance states during the light period. All mice were subjected to at least two recording sessions with at least a 2 day interval in between each session to allow photobleaching recovery. We separated all sleep-state transitions that last at least for 1 min before and after the state change. All the sessions were selected after the photometry signal became stable, as we observed a decay of photometry signal at the beginning of the recordings.

## Sleep deprivation

Mice expressing ACR2-2A-mCherry in VTA$_{Gad67+}$ neurons were used for optogenetic inhibition in *Figure 3* and were subjected to sleep deprivation. Mice were submitted to complete sleep deprivation, through the gentle handling method, which consists of keeping the animal awake by gently touching them with a soft brush if behavioral signs of sleep were observed. Mice were sleep-deprived for 4 hr, starting at light onset. Mice were then allowed to experience recovery sleep for 30 min. Optogenetic inhibition was performed during the recovery sleep.

## Acute brain slice preparation

Preparation of acute brain slices and subsequent electrophysiological recordings were performed as previously reported with a slight modification (*Chowdhury and Yamanaka, 2016*). Briefly, mice were decapitated under isoflurane (Fujifilm Wako Pure Chemical Industries) anesthesia and the brain was quickly isolated and chilled in an ice-cold cutting solution (in mM: 110 K-gluconate, 15 KCl, 0.05 EGTA, 5 HEPES, 26.2 NaHCO$_3$, 25 glucose, 3.3 MgCl$_2$ and 0.0015 (±)−3-(2-carboxypiperazin-4-yl) propyl-1-phosphonic acid) gassed with 95% O$_2$ and 5% CO$_2$. Coronal slices of 300 μm thickness containing either VTA or LH were prepared using a vibratome (VT-1200S; Leica, Wetzlar, Germany) and were temporarily placed in an incubation chamber containing a bath solution (in mM: 124 NaCl, 3 KCl, 2 MgCl$_2$, 2 CaCl$_2$, 1.23 NaH$_2$PO$_4$, 26 NaHCO$_3$ and 25 glucose) gassed with 95% O$_2$ and 5% CO$_2$ in a 35°C water bath for 30–60 min. Slices were then incubated at room temperature in the same incubation chamber for another 30–60 min for recovery.

## In vitro electrophysiology

After the recovery period, acute brain slices were transferred to a recording chamber (RC-26G; Warner Instruments, Hamden, CT). The recording chamber was equipped with an upright

fluorescence microscope (BX51WI; Olympus, Tokyo, Japan) stage and was superfused with a 95% $O_2$ and 5% $CO_2$-gassed bath solution at a rate of 1.5 ml/min using a peristaltic pump (Dynamax; Rainin, Oakland, CA). An infrared camera (C3077-78; Hamamatsu Photonics, Hamamatsu, Japan) was installed in the fluorescence microscope along with an electron multiplying charge-coupled device camera (Evolve 512 delta; Photometrics, Tucson, AZ) and both images were separately displayed on monitors. Micropipettes of 4–6 MΩ resistance were prepared from borosilicate glass capillaries (GC150-10; Harvard Apparatus, Cambridge, MA) using a horizontal puller (P-1000; Sutter Instrument, Novato, CA). Patch pipettes were filled with KCl-based internal solution (in mM: 145 KCl, 1 $MgCl_2$, 10 HEPES, 1.1 EGTA, 2-Mg-ATP, 0.5 $Na_2$-GTP; pH 7.3 with KOH) with osmolality between 280–290 mOsm. Electrophysiological properties of cells were monitored using an Axopatch 200B amplifier (Axon Instrument, Molecular Devices, Sunnyvale, CA). Output signals were low-pass filtered at 5 kHz and digitized at a sampling rate of 10 kHz. Patch clamp data were recorded through an analog-to-digital (AD) converter (Digidata 1550A; Molecular Devices) using pClamp 10.2 software (Molecular Devices). Voltage clamp recordings were performed at a holding potential of $-60$ mV, unless otherwise stated. Blue light at a wavelength of 475 ± 18 nm and yellow light at a wavelength of 575 ± 13 nm were generated by a light source that used a light-emitting diode (Spectra Light Engine; Lumencor, Beaverton, OR) and guided to the microscope stage with a 1 cm diameter optical fiber. Brain slices were illuminated through the objective lens of the fluorescence microscope.

### In vitro calcium imaging

Gad67+ neurons were identified via green fluorescence of GCaMP6f. Excitation light of 475 ± 18 nm (6.8 mW/mm$^2$) was emitted into the brain slice containing VTA through the objective lens of a fluorescence microscope. The light source (Spectra light engine) was controlled by Metamorph software (Molecular Devices). GCaMP6f fluorescence intensity was recorded continuously using Metamorph software at a rate of 1 Hz with 100 ms of exposure time. To synchronize calcium imaging and patch clamp recording, pClamp software was triggered by the TTL output from Metamorph software. Metamorph data were analyzed by setting the region of interest (ROI) on GCaMP6f-expressing VTA$_{Gad67+}$ neurons and the ΔF/F was calculated from the average intensity of the ROI. Finally, ΔF/F values for 10, 20, and 50 Hz were normalized to the ΔF/F values for corresponding 100 Hz frequencies.

### Data analysis and presentation

Immunostaining data were analyzed and processed with ImageJ (US National Institute of Health) and BZ-X Analyzer (Keyence BZ-X710 microscope). Electrophysiological analysis was performed with either Clampfit10 (Molecular Devices, Sunnyvale, CA) or Minianalysis software (Synaptosoft Inc, Decatur, GA). Analysis of EEG-EMG data was performed using SleepSign software (Kissei Comtec) and data were outputted as text files. Further analyses were performed using Microsoft Excel. Electrophysiological data were saved as American Standard Code for Information Interchange (ASCII) files and further data calculations were performed in Microsoft Excel. Graphs were generated in Origin 2017 (OriginLab, Northampton, MA) using data from Excel. Statistical analysis was also performed with Origin 2017. Graphs were generated using Canvas 15 (ACD Systems, Seattle, WA).

## Acknowledgements

We thank CJ Hung and G Wang for supporting the sleep data analysis, SM Rahaman for assisting cell counting procedures, S Kikuchi for sleep deprivation and S Tsukamoto and Y Miyoshi for other technical assistance.

## Additional information

### Funding

| Funder | Grant reference number | Author |
| --- | --- | --- |
| Japan Science and Technology Agency | JPMJCR1656 | Akihiro Yamanaka |

| Ministry of Education, Culture, Sports, Science, and Technology | 17H05563 | Akihiro Yamanaka |
|---|---|---|
| Ministry of Education, Culture, Sports, Science and Technology | 18KK0223 | Akihiro Yamanaka |
| Ministry of Education, Culture, Sports, Science and Technology | 18H05124 | Akihiro Yamanaka |
| Ministry of Education, Culture, Sports, Science and Technology | 16H01271 | Akihiro Yamanaka |
| Ministry of Education, Culture, Sports, Science and Technology | 18H02523 | Akihiro Yamanaka |
| Ministry of Education, Culture, Sports, Science and Technology | 19H05016 | Akihiro Yamanaka |

The funders had no role in study design, data collection and interpretation, or the decision to submit the work for publication.

## Author contributions

Srikanta Chowdhury, Conceptualization, Data curation, Formal analysis, Writing—original draft, Writing—review and editing; Takanori Matsubara, Toh Miyazaki, Daisuke Ono, Noriaki Fukatsu, Data curation, Formal analysis; Manabu Abe, Kenji Sakimura, Yuki Sudo, Resources; Akihiro Yamanaka, Conceptualization, Supervision, Funding acquisition, Writing—review and editing

## Author ORCIDs

Srikanta Chowdhury (iD) https://orcid.org/0000-0002-2216-5960
Akihiro Yamanaka (iD) https://orcid.org/0000-0001-6099-7306

## Ethics

Animal experimentation: All experimental protocols that involved animals were approved by the Institutional Animal Care and Use Committees, Research Institute of Environmental Medicine, Nagoya University, Japan. All efforts were made to reduce the number of animals used and also to minimize the suffering and pain of animals.(Approval number #18232, #18239).

## Decision letter and Author response

Decision letter https://doi.org/10.7554/eLife.44928.sa1
Author response https://doi.org/10.7554/eLife.44928.sa2

# Additional files

## Supplementary files

• Transparent reporting form

## Data availability

All data generated or analysed during this study are included in the manuscript and supporting files. Source data files have been provided for Figures 1-8 and Figure Supplements.

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
