## [Decision Letter]

Thank you for submitting your article "GABA neurons in the ventral tegmental area regulate non-rapid eye movement sleep in mice" for consideration by *eLife*. Your article has been reviewed by three peer reviewers, including Yang Dan as the Reviewing Editor and Reviewer #1, and the evaluation has been overseen by Catherine Dulac as the Senior Editor.

The reviewers have discussed the reviews with one another and the Reviewing Editor has drafted this decision to help you prepare a revised submission.

Summary:

In this study, the authors investigated the effect of GAD67-positive GABAergic neurons in the VTA in regulating sleep. They showed that chemogenetic activation of these neurons promote NREM sleep and optogenetic inhibition of them promptly awakens the mice from NREM but not REM sleep. Fiber photometry showed that these GABAergic neurons show highest activity during NREM sleep. Slice recording showed that these neurons provide GABAergic inhibition of orexin neurons, a pathway that could contribute to the sleep promoting effect.

Essential revisions:

An important difference between this study and a recently published study (Yu et al.) is the activity profile of the VTA GABAergic neurons. The authors discussed that notable difference is the use of VGAT-Cre vs. GAD67-Cre, but this point needs to be strengthened. The counting results of overlap between hrGFP and *Gad67* is not provided, and there seems to be hrGFP+; *Gad67*- cells in Figure 1B. Perhaps this is because *Gad67* IHC does not give a clear image of the soma in VTA. Please conduct double staining of *Gad67* ISH and hrGFP IHC and provide the counting data. In addition, as some neurons express *Gad67* but not *Vgat*, please also conduct double staining of *Vgat* ISH and hrGFP IHC and provide the counting data. In each case, please provide both (Marker gene+ & hrGFP+ cells)/(marker gene+ cells) and (Marker gene+ & hrGFP+ cells)/(hrGFP+ cells). Please also provide number of mice used for each analysis in the figure legend.

Other points:

Figure 1: legends should list abbreviations alphabetically.

Results, subsection “GABAergic neurons in the VTA project to brain areas involved in the regulation of sleep/wakefulness”:

“We counted a total of 636 ± 122 neurons per animal.” Are these serial sections with all neurons counted? Unilateral or bilateral? Thickness of sections? Only cells with visible nucleus, or all labelled cells?

Define DAPI in the Figure 1 legend. Explain why it was used (presumably to label DNA (RNA?) in all cells?)

Figure 2: the changes in sleep are indeed impressive.

Figure 5 nicely shows activation of Gad67 neurons in NREM.

Figures 6 and 7: Chemelli and others have shown that narcoleptic mice do not actually have much more sleep than WT. They differ in having few long duration sleep episodes, as do human narcoleptics. This is relevant to the authors stressing the orexin role. It is important to appreciate that removal of orexin neurons causing narcolepsy disrupts both waking and sleep. So the implication that orexin is mediating the VTA Gad67 effects is not impressively supported, despite these nice in vitro studies.

It would be desirable to study the long-term effects of removal of the VTA_Gad67_ neurons, as done by Shank et al. Would there be a permanent effect on sleep?

The authors claim that "VTA_Gad67+_ neurons play a crucial role in the regulation of NREM sleep." They assert that arousal is "critically regulated" by the VTA. What they actually demonstrate is more modest. Like most brain neurons, their activation or inhibition can have some effect on sleep. They do not demonstrate long-term effects, only sleep interruption or initiation effects. Manipulation of many brain areas in sleep will produce waking. Also, there is an extensive literature on the effects of pharmacologic manipulation of noradrenergic, serotonergic, histaminergic and cholinergic neurons on sleep. Are all of these "crucial?" If so, the word begins to lose its meaning. So, they should put their interesting work in the context of other work on basal forebrain, hypothalamic and brainstem contributions to sleep, beyond the limited context they already provide.

Introduction: "However, no study has been conducted to date to confirm the roles of GABAergic neurons in the VTA in the regulation of sleep/wakefulness"

This is incorrect. Two papers have confirmed the involvement of the neurons in sleep regulation.

Results, subsection “VTA_Gad67+_ neurons directly inhibited wake-promoting orexin neurons in the lateral hypothalamus”: It is difficult to understand the underlying logic of why the combined results of TTX and 4-AP lead to a conclusion that VTA GABAergic neurons form monosynaptic connection with LH orexin neurons. Please provide more detailed explanation here.

Figure 2E: Please also provide EEG spectrum of wake and REMS.

Figure 2G: Please state in the figure legend whether this figure indicates delta power specifically during NREMS or during all states.

Please state in the figure legends the number of mice used for each group.

Figure 4: The method for sleep deprivation is not clear. Please describe in the text.

Figure 5—figure supplement 1: The EEG pattern seems inconsistent between panels. Especially, the REMS in the REMS to wake transition looks more like NREMS. Why is this?

Since recently there are so many subtypes of optogenetic tools, please precisely define the type of opsins used (for example, by providing Genbank ID).

---

## [Author Response]

Essential revisions:An important difference between this study and a recently published study (Yu et al.) is the activity profile of the VTA GABAergic neurons. The authors discussed that notable difference is the use of VGAT-Cre vs. GAD67-Cre, but this point needs to be strengthened. The counting results of overlap between hrGFP and Gad67 is not provided, and there seems to be hrGFP+; Gad67- cells in Figure 1B. Perhaps this is because Gad67 IHC does not give a clear image of the soma in VTA. Please conduct double staining of Gad67 ISH and hrGFP IHC and provide the counting data. In addition, as some neurons express Gad67 but not Vgat, please also conduct double staining of Vgat ISH and hrGFP IHC and provide the counting data. In each case, please provide both (Marker gene+ & hrGFP+ cells)/(marker gene+ cells) and (Marker gene+ & hrGFP+ cells)/(hrGFP+ cells). Please also provide number of mice used for each analysis in the figure legend.

We agree with this point. To address this, we performed in situ RNA hybridization assays using RNAscope technology (added as Figure 1—figure supplement 1). We injected Cre-inducible AAV (AAV(9) CAG-FLEX-mCherry) into the VTA of Gad67-Cre mice (n = 4 mice) to label Gad67-expressing neurons (Figure 1—figure supplement 1A). *Gad67, Vgat* (vesicular GABA transporter), and *mCherry* mRNA were visualized by multicolor in situ hybridization (Figure 1—figure supplement 1B).

Regarding *Gad67* and *Vgat* expression, we revealed three different types of neurons: *Vgat*-only (66.8 ± 2.0%), *Gad67*-only (1.1 ± 0.1%), and *Vgat* and *Gad67* double-positive neurons (32.1 ± 2.1%) (Figure 1—figure supplement 1C). Whereas 96.5 ± 0.5% of *Gad67*-positive neurons co-expressed *Vgat* mRNA, only 32.4 ± 2.1% of *Vgat*-positive neurons co-expressed *Gad67* mRNA in the VTA (Figure 1—figure supplement 1C). Consistent with these findings, 94.8 ± 1.5% of mCherry-positive neurons in the VTA of *Gad67-Cre* mice co-expressed *Gad67* and *Vgat* mRNA (Figure 1—figure supplement 1D). Together, these data clearly indicate that *Gad67*-positive neurons represent a small subset of *Vgat*-positive neurons in the VTA. Therefore, this data well supports our assumption that different populations of GABAergic neurons were labelled in Gad67-Cremice and Vgat-cre mice in the VTA. This difference might result in alterations of in vivo activity profiles across vigilance states recorded by fiber photometry.

Other points:Figure 1: legends should list abbreviations alphabetically.

Abbreviations are now listed alphabetically in the legends of the revised manuscript.

Results, subsection “GABAergic neurons in the VTA project to brain areas involved in the regulation of sleep/wakefulness”:“We counted a total of 636 ± 122 neurons per animal.” Are these serial sections with all neurons counted? Unilateral or bilateral? Thickness of sections? Only cells with visible nucleus, or all labelled cells?

The revised manuscript includes all of the above information. We added the following sentence to the Results section: “At least 3 weeks after unilateral injection of AAV aimed at the VTA to express hrGFP, we prepared coronal brain sections at 40 µm thickness and counted all labeled cells on every fourth section.”

Define DAPI in the Figure 1 legend. Explain why it was used (presumably to label DNA (RNA?) in all cells?)

We have included an explanation about DAPI in the Figure 1 legend of the revised manuscript. DAPI staining was used to label nuclear DNA and also to assist understanding of the anatomical position of VTA_Gad67+_ neuronal projections.

Figure 2: the changes in sleep are indeed impressive.Figure 5 nicely shows activation of Gad67 neurons in NREM.

We thank the reviewer for their appreciation of our data.

Figures 6 and 7: Chemelli and others have shown that narcoleptic mice do not actually have much more sleep than WT. They differ in having few long duration sleep episodes, as do human narcoleptics. This is relevant to the authors stressing the orexin role. It is important to appreciate that removal of orexin neurons causing narcolepsy disrupts both waking and sleep. So the implication that orexin is mediating the VTA Gad67 effects is not impressively supported, despite these nice in vitro studies.

In vitro experiments in this study showed that LH_orexin_ neurons were directly innervated and inhibited by VTA_Gad67+_ neurons. To further support the implication that VTA_Gad67+_ neurons regulate sleep/wakefulness through orexin neurons in mice, we performed projection-specific optogenetic activation of VTA_Gad67+_ neurons in the LH (Figure 8). In vivo studies suggest that the VTA_Gad67+_ neurons promote NREM sleep, at least in part, through their projection to the LH. We added the following text to the Results section “VTA_Gad67+_ neurons mediate a sleep-promoting function via the lateral hypothalamus”:

“Finally, to test whether the VTA_Gad67+_ to LH projection participates in the induction of NREM sleep, we performed in vivo optogenetic activation of VTA_Gad67+_ nerve terminals in the LH. […] These results suggest that VTA_Gad67+_ neurons promote NREM sleep, at least in part, through their projection to the LH.”

It would be desirable to study the long-term effects of removal of the VTA_Gad67_ neurons, as done by Shank et al. Would there be a permanent effect on sleep?

While designing the study, we intended to determine the long-term effects of removal of the VTA_Gad67_ neurons. Aiming at the VTA, we performed bilateral intracranial microinjection of AAV in the Gad67-Cre mice to express diphtheria toxin A fragment (DTA) in the VTA_Gad67+_ neurons. However, to our surprise, we found that all injected mice (n = 3 male mice) died within 4 weeks of injection. Although we cannot conclude the exact reasons for such mortality of the AAV-injected Gad67-Cre mice, we did not continue this line of experimentation to investigate the long-term effects of removal of VTA Gad67 neurons due to ethical concerns.

The authors claim that "VTA_Gad67+_ neurons play a crucial role in the regulation of NREM sleep." They assert that arousal is "critically regulated" by the VTA. What they actually demonstrate is more modest. Like most brain neurons, their activation or inhibition can have some effect on sleep. They do not demonstrate long-term effects, only sleep interruption or initiation effects. Manipulation of many brain areas in sleep will produce waking. Also, there is an extensive literature on the effects of pharmacologic manipulation of noradrenergic, serotonergic, histaminergic and cholinergic neurons on sleep. Are all of these "crucial?" If so, the word begins to lose its meaning. So, they should put their interesting work in the context of other work on basal forebrain, hypothalamic and brainstem contributions to sleep, beyond the limited context they already provide.

We replaced the word “crucial” with “important”. We also discussed the role of VTA_Gad67+_ neurons in the context of other works in the Discussion section of the revised manuscript. We added the following sentences: “Moreover, while many brain areas including the cholinergic basal forebrain and brain stem, histaminergic posterior hypothalamus, serotonergic raphe nucleus, as well as the noradrenergic locus coeruleus are important in sleep/wake regulation (Brown et al., 2012), we speculate a critical role for VTA_Gad67+_ neurons in sleep/wakefulness-regulating circuitry as these neurons modulate diverse targets, including DA neurons in the VTA (Tan et al., 2012)”.

Introduction: "However, no study has been conducted to date to confirm the roles of GABAergic neurons in the VTA in the regulation of sleep/wakefulness"This is incorrect. Two papers have confirmed the involvement of the neurons in sleep regulation.

We agree with the reviewer’s comment. This sentence was deleted from the revised manuscript.

Results, subsection “VTA_Gad67+_ neurons directly inhibited wake-promoting orexin neurons in the lateral hypothalamus”: It is difficult to understand the underlying logic of why the combined results of TTX and 4-AP lead to a conclusion that VTA GABAergic neurons form monosynaptic connection with LH orexin neurons. Please provide more detailed explanation here.

We now provide a detailed explanation and reference as to why the combined results of TTX and 4-AP lead to a conclusion that VTA GABAergic neurons form monosynaptic connections with LH orexin neurons in the revised manuscript. We added the following description in the Results section: “However, combined application of TTX along with 4-aminopyridine (4-AP, 1 mM), a voltage-gated potassium (Kv) channel blocker that prolongs depolarization of axon terminals and enables ChR2-mediated release of neurotransmitter in the absence of action potentials (Petreanu et al., 2009), could rescue the light-induced PSCs, […]”

Figure 2E: Please also provide EEG spectrum of wake and REMS.

We have now provided the EEG power spectrums for wake and REMS as a new supplementary figure (Figure 2—figure supplement 2). Because mice show little to no wake and REM sleep at 1 hr post-CNO injection, we showed the EEG power spectrum of wake and REM sleep as “Pre” and “4 hr Post”. Note that the EEG of the wake at 4 hr post-CNO injection shows an EEG power spectrum with lower theta power, presumably because pharmacological activation effects last up to 4 hours after CNO i.p. administration.

Figure 2G: Please state in the figure legend whether this figure indicates delta power specifically during NREMS or during all states.

Figure 2G indicates delta power specifically during NREM sleep, and this information is now indicated in the figure legend of the revised manuscript.

Please state in the figure legends the number of mice used for each group.

The number of mice in each group has now been indicated in all the figure legends of the revised manuscript.

Figure 4: The method for sleep deprivation is not clear. Please describe in the text.

Our sleep deprivation protocol is now indicated in the Materials and methods section of the revised manuscript. We added the following sentences: “Sleep deprivation Mice expressing ACR2-2A-mCherry in VTA_Gad67+_ neurons were used for optogenetic inhibition in Figure 3 and were subjected to sleep deprivation. Mice were submitted to complete sleep deprivation, through the gentle handling method, which consists of keeping the animal awake by gently touching them with a soft brush if behavioural signs of sleep were observed. Mice were sleep-deprived for 4 hr, starting at light onset. Mice were then allowed to experience recovery sleep for 30 min. Optogenetic inhibition was performed during the recovery sleep”

Figure 5—figure supplement 1: The EEG pattern seems inconsistent between panels. Especially, the REMS in the REMS to wake transition looks more like NREMS. Why is this?

We thank the reviewer for noting this. We agree that the EEG pattern of the REM sleep in the REMS-to-wake transition looks more like NREMS, although considering other parameters this stage has been determined as REM sleep. Because of such ambiguity in the EEG pattern, we have replaced the trace in the REM-to-wake transition in Figure 5—figure supplement 1.

Since recently there are so many subtypes of optogenetic tools, please precisely define the type of opsins used (for example, by providing Genbank ID).

We have now provided the Genbank accession number of anion channelrhodopsin-2 (ACR2), accession number KP171709, as well as cited the original papers for both ACR2 and ChR2(E123T/T159C) in the revised version of the manuscript.